# Identification of Differential N-Glycan Compositions in the Serum and Tissue of Colon Cancer Patients by Mass Spectrometry

**DOI:** 10.3390/biology10040343

**Published:** 2021-04-20

**Authors:** Marcelo de M.A. Coura, Eder A. Barbosa, Guilherme D. Brand, Carlos Bloch, Joao B. de Sousa

**Affiliations:** 1Division of Colorectal Surgery, University Hospital of Brasilia, School of Medicine, University of Brasilia, SGAN 605, Brasilia-DF 70840-901, Brazil; sousajb@unb.br; 2Laboratory of Mass Spectrometry, EMBRAPA Genetic Resources and Biotechnology, Parque Estação Biológica, PqEB, Av. W5 Norte, Brasilia-DF 70770-917, Brazil; bioederr@gmail.com (E.A.B.); carlos.bloch@embrapa.br (C.B.J.); 3Laboratory for the Synthesis and Analysis of Biomolecules, Institute of Chemistry, Campus Universitario Darcy Ribeiro, University of Brasilia, Brasilia-DF 70910-900, Brazil; gdbrand@gmail.com

**Keywords:** colorectal cancer, LC/MS, MALDI-TOF/MS, N-glycosylation, mass spectrometry

## Abstract

**Simple Summary:**

Incidence of colorectal cancer (CRC) has been rising in Brazil. To date, no reliable biomarker has been described in CRC for diagnosis and prognosis. Modifications in the N-glycosylation profile are usually associated with many cancers, as CRC. In turn, mass spectrometry (MS)-based methods are the most accurate technology in quantification of N-glycans. Therefore, we described a unique pattern of compositions altered in serum and tissues of stages II and III colon cancer patients, identified by MALDI-TOF/MS and LC-MS technology. N-glycans were mostly found decreased in serum whilst oligomannosidic, hypogalactosylated, and tetra-antennary forms were overexpressed in tumor tissues. Total N-glycome in serum of cancer patients was different from the profile found in serum of healthy individuals. Strikingly, no correlation between tissue N-glycosylation profile and serum profile was observed in cancer patients, posing the question where these compositions are originated from.

**Abstract:**

Colorectal cancer (CRC) ranks second as the leading cause of cancer-related deaths worldwide. N-glycosylation is one of the most common posttranslational protein modifications. Therefore, we studied the total serum N-glycome (TSNG) of 13 colon cancer patients compared to healthy controls using MALDI-TOF/MS and LC-MS. N-glycosylation of cancer tumor samples from the same cohort were further quantified using a similar methodology. In total, 23 N-glycan compositions were down-regulated in the serum of colon cancer patients, mostly galactosylated forms whilst the mannose-rich HexNAc2Hex7, the fucosylated bi-antennary glycan HexNAc4Hex5Fuc1NeuAc2, and the tetra-antennary HexNAc6Hex7NeuAc3 were up-regulated in serum. Hierarchical clustering analysis of TSNG correctly singled out 85% of the patients from controls. Albeit heterogenous, N-glycosylation of tumor samples showed overrepresented oligomannosidic, bi-antennary hypogalactosylated, and branched compositions related to normal colonic tissue, in both MALDI-TOF/MS and LC-MS analysis. Moreover, compositions found upregulated in tumor tissue were mostly uncorrelated to compositions in serum of cancer patients. Mass spectrometry-based N-glycan profiling in serum shows potential in the discrimination of patients from healthy controls. However, the compositions profile in serum showed no parallel with N-glycans in tumor microenvironment, which suggests a different origin of compositions found in serum of cancer patients.

## 1. Introduction

Colorectal cancer (CRC) is the third most common cancer worldwide, with an annual incidence of approximately 1.8 million new cases, of which 900,000 are expected to die from this malignancy [1]. While its incidence is declining in developed countries, a steady increase has been observed in Brazil in recent years [2].

Survival is closely related to clinical stage as fewer than 10% patients with metastatic disease will survive longer than 5 years. On the other hand, patients with early diagnosis show a disease-free survival rate close to 90% [3]. Therefore, early detection of CRC is key. Currently, colonoscopy is the gold standard for diagnostic purposes in CRC, and albeit highly accurate, this is an invasive method, not exempt from complications. In turn, many biomarkers currently in use, mostly carbohydrate antigens such as CEA and CA 19-9, show low sensitivity and specificity in CRC early detection [4].

After surgical resection, surveillance is recommended for CRC patients to early detection of recurrent disease. Even though many patients will receive adjuvant chemotherapy, some of them will not benefit from treatment, eventually developing metastasis. Therefore, there is a pressing need to discover new and more reliable biomarkers for early detection and prognosis in CRC, maybe present in human tissues or in biofluids [5,6].

Selective pressure within the tumor microenvironment leads to significant modifications of several clones, with better adapted tumor cells overgrowing. Progressive mutations in oncogenes and tumor suppressor genes enable malignant cells to cope with these environment changes [7].

However, more plastic phenotype expressions, as epigenetic and posttranslational events, are frequently involved in tumor development. In fact, cell-cell, cell-extracellular matrix, and complex interplay between host immune system and tumor cells are all dynamic processes, highly mediated by glycocalyx interaction [8].

Glycosylation is one of the most common posttranslational modifications (PTM) in proteins where oligosaccharides are covalently attached to nascent polypeptide chains in the endoplasmic reticulum. These carbohydrates are also involved in proper protein folding, final intracellular location, or secretion as extracellular vesicles, turnover, and proteolysis. Approximately 60% of all human proteins are glycosylated, with N-glycosylation occurring when a glycan is attached to asparagine (Asn) in the motif Asn-X-Ser/Thr, where X is not a proline residue [9].

Unlike DNA or proteins, glycans are formed with no previous template, which increases the complexity and possibility of variations in compositions and structures. Therefore, high-throughput analytic technologies, such as capillary electrophoresis, high and ultra-performance liquid chromatography, and modern mass spectrometry (MS), are needed to perform in-depth analysis of micro-structural composition and quantification of these highly variable structures (e.g., glycoconjugates) that are usually present in minute amounts into biologic materials [10,11]. 

Aberrant glycosylation has been acknowledged as a hallmark of cancer as a result of alterations in machinery of tumor cells, albeit it is not clear this phenotypical modification is cause of consequence of malignant transformation [7,12,13]. Over the last 10 years, many studies have focused on the role of N-glycosylation in CRC. Albeit heterogeneous, colorectal tumors revealed a distinct pattern including increase of mannose rich, acidic N-glycans, core-fucosylated bi-antennary, multi-antennary sialylated and decrease of bisected forms in microenvironment of tumor samples [14,15,16,17,18,19,20].

Recently, mass spectrometry imaging (MSI)-based study in stage II CRC has revealed spatial distribution of N-glycans in the tumor microenvironment with increase of mannose rich and sialylation levels in stroma, spreading into the tumor invasive front [21].

Since tumors may shed glycoproteins into serum, the analysis of N-glycome in blood samples from cancer patients may serve as a read out of the tumor microenvironment, therefore, providing a reliable source of cancer biomarkers [22]. 

More importantly, the human total serum N-glycome (TSNG) has comprehensively been described and, albeit variations amongst normal individuals, it is believed that many pathologic states including inflammation and cancer promote significant modifications on glycoproteins, easily identifiable from healthy individuals [23,24,25].

In this setting, by using fluorophore-assisted carbohydrate electrophoresis (FACE), Zhao et al. were able to find globally decreased levels of core-fucosylated N-glycans in serum of colorectal cancer patients with high predictive value in single out cases from adenoma patients and healthy individuals [26].

Relatedly, Doherty et al. analyzed by Ultra-High Performance Liquid Chromatography (UHPLC) technology the TSNG of a large cohort of CRC patients and controls. The authors observed overall decrease in core-fucosylated neutral biantennary N-glycans, whilst chromatograms peaks of multi-antennae sialylated glycoforms were found increased in serum of CRC cases [27].

The present work also deals with similar CRC events as mentioned on the studies above. Nevertheless, previous MS-based studies for biomarker discovery have addressed the N-glycome in serum or the N-glycome in tissues of CRC patients separately, as one assumes that compositions found in blood stream may reflect glycosylation present in the tumor microenvironment. In this study we analyzed, both serum and tissue N-glycome of CRC patients, by two MS-based technologies, MALDI-TOF/MS and LC-MS, which revealed a special serum pattern compared to healthy individuals. More importantly, N-glycome in serum of CRC patients was highly different from compositions observed in tumor tissues.

## 2. Materials and Methods

### 2.1. Chemical and Reagents

Guanidine chloride, glacial acetic acid, formic acid (FA), and chloroform (Bio Clinical Lab, Phillipsburg, NJ, USA). Dimethyl sulfoxide (DMSO), sodium hydroxide beads, iodomethane, methyl-deuterium iodide, Trypsin (TPCK-treated from bovine pancreas) were purchased from Sigma-Aldrich (St. Louis, MO, USA). N-Glycanase (PNGase F) was purchased from Prozyme (Hayward, CA, USA). Meanwhile, MALDI-TOF/MS calibrant mixture (calibrant Standard II) 2,5-Dihydroxybenzoic acid (2,5-DHB) was purchased from Bruker Daltonics (Bremen, Germany). LC-MS grade water, acetonitrile, and methanol were purchased from Sigma-Aldrich (St. Louis, MO, USA). Sep-pak C-18 E 3.0 cc cartridges Strata X SPE were purchased from Phenomenex (Torrance, CA, USA), while Amicon Ultra-0.5 Centrifugal Filter Unit with Ultracel-10 membrane was from EMD Millipore (Billerica, MA, USA).

### 2.2. Patients and Controls

We evaluated 15 patients with sporadic microsatellite stable (MSS) colon cancer in stage II or III disease, who underwent surgery at the University Hospital of Brasilia from October 2017 to June 2018.

Initially, sporadic cancer was defined as cases that not fulfilled Amsterdam II criteria. Synchronous or metachronous tumors, tumors from polyposis adenomatous familial phenotype, and tumors in the inflammatory bowel disease (IBD) setting were not included.

To rule out Lynch syndrome and other CRC hereditary syndromes, all enrolled patients underwent 37-panel genetic testing to detect germinative pathogenic mutations (APC ATM BARD1 BLM BRCA1 BRCA2 BRIP1 CDH1 CDK4 CDKN2A CHEK2 EGFR EPCAM FANCC MEN1 MET MLH1 MSH2 MSH6 MUTYH NBN NF1 NF2 PALB2 PIK3CA PMS2 PTEN RAD51C RAD51D RB1 RECQL RET STK11 TP53 WT1 POLD1 POLE).

All tumor samples demonstrated proficient DNA mismatch repair protein expression on immunohistochemistry analysis. Each tumor sample was also tested for EGFR, NRAS, KRAS (codons 12 and 13), and BRAF ^V600E^ hot spot mutation in exon 15.

Staging was according to the 8th edition of American Joint of Committee on Cancer [28].

For the control group, we enrolled 15 never smokers, non-diabetic, with no-inflammatory chronic conditions, otherwise healthy individuals. They had screening colonoscopies in which no inflammatory bowel disease (IBD), polyps, or cancer were detected.

This study was approved by the Ethic Committee and Research of University of Brasilia, UnB DOC 63183716.0.0000.5558/2017. Written informed consent was created in accordance with Declaration of Helsinki and voluntarily obtained from all patients and controls.

### 2.3. Samples

#### 2.3.1. Serum

Blood samples (5 mL) from each colon cancer patient were collected in the morning before the surgical procedure plus from 11 healthy controls, immediately before colonoscopy. After centrifugation for 4 min at 13,400 rpm, serum was collected and frozen at −80 °C.

#### 2.3.2. Tissue

Immediately after the resection of the surgical specimen, 6 fragments 1 × 1 cm were collected: 2 from the proximal tumor border, 2 from the distal tumor border and, 2 fragments from mucosa and submucosa of macroscopically normal colonic tissue, located 10 cm distant from the tumor border. All 6 fragments were frozen at −80 °C.

All tumor samples demonstrated proficient DNA mismatch repair protein expression (MSS tumors), identified by immunohistochemistry analysis. Each tumor sample was also tested for EGFR, NRAS, KRAS (codons 12 and 13), and BRAF V600E mutations. Only distal fragments were processed for tissue extraction. Proximal samples were stored as repository for further analysis when deemed necessary.

### 2.4. Protein Extraction from Tissue Samples

Tissues were pulverized on liquid nitrogen by using a mortar and were stored at −80 °C. For protein extractions, 60 mg of powdered tissue were dissolved in 500 µL of a 50% acetonitrile (ACN) solution in water, vortexed for 1 min, extracted in a bath sonicator for 45 min at room temperature, and subsequently centrifuged at 14,000 rpm for 45 min. The upper liquid phase was collected and freeze dried for posterior glycoprotein extraction.

### 2.5. N-Glycan Isolation and Derivatization

Isolation of N-linked oligosaccharides was carried out by using the method described by Morelle and Michalsky [29] with some minor modifications. Briefly, 40 µL of serum or all powdered tissue extract were submitted to protein reduction and alkylation. For reduction, it was used 9.6 μL of dithiothreitol (DTT) 500 mM and for alkylation 9.2 μL iodoacetoamide (IAA) 3M, in 200 μL of a buffer solution (pH 8.4). Polypeptides were filtered through size-exclusion Centricon, with 10 kDa cut-off and posteriorly submitted to trypsin proteolysis. Trypsin digestion was carried out by adding 14 μL of a trypsin solution (5 μg/μL) to reduced and alkylated glycoproteins for 24 h, at 37 °C. N-glycans were released from the digested glycoprotein after incubation with 3 μL of PNGAse F solution (1 μg/μL) at 37 °C for 17 h, and these were subsequently purified by sequential elution in 5% acetic acid through a Sep-Pak C18 column. Sep-Pak C18 columns were sequentially conditioned with 5 mL of methanol and 10 mL of 5% acid acetic solution. Samples containing free N-glycans and peptides were resuspended in 200 μL of 5% acetic acid and added to the previously conditioned Sep-Pak columns. Sequential elutions were carried out by adding 3 mL of 5% acetic acid and 3 mL of 5% acetic acid with 80% acetonitrile. Free N-glycans were collected after elution with 3 mL of 5% acetic acid. N-glycans isolated from colorectal cancer tissues and serum of CRC patients were derivatized using a light isotope of iodomethane (I-CH_3_) while N-glycans isolated from normal colorectal tissues and serum of control patients were derivatized using the heavy isotope of iodomethane (deuterated, I-CD_3_), under continuous N_2_ flushing. Derivatized N-glycans were extracted by a liquid/liquid approach using chloroform that was repeatedly washed with Milli-Q water and dried by a stream of nitrogen. Samples were dissolved in acetonitrile and subsequently purified using a Sep-Pak C18: derivatized N-glycans eluted from Sep-Pack column in 80% acetonitrile after performing washes with water and 10% acetonitrile solution, consecutively. Purified derivatized N-glycans were collected and freeze dried.

### 2.6. Determination of N-Glycan Profile from Blood Samples

The profile of N-glycans from blood samples was performed like described before [30], with minor modifications. Briefly, the same volume of the blood plasma obtained from control patients (11 samples) were joined in a solution denominated control pool. N-glycans isolated from each individual (13 colon cancer and 11 control patients) were derivatized with the light isotope of iodomethane (I-CH_3_) while 24 samples corresponding to N-glycans isolated from control pool were derivatized using a heavy isotope of iodomethane (I-CD_3_). After purification of the derivatized N-glycans, each of the permethylated samples were mixed with another sample corresponding to a control pool, in a proportion of 1:1. Samples were freeze dried and stored at −80 °C before analysis. All 24 samples, containing methylated and deuterated glycans were analyzed by MALDI-TOF/MS and LC-MS. The ratio between signals produced in mass spectra or TIC for each methylated glycan was calculated by using the heavy isotope signal corresponding to the same structure, when present, as reference (I-CH_3_/I-CD_3_ signals). Choice of the heavy reference for structures without corresponding deuterated signal was based on both structural similarity and signal intensity (Appendix A). Additionally, the values of the ratios obtained for each N-glycan structure were normalized by the area ratio of the bi-antennary glycan HexNAc4Hex5NeuAc2 ([M + Na]^+^ = 2792.4 Da), and this glycan (52 in our numbering system) was excluded from further analyses. Normalized area ratios were standardized and submitted to hierarchical clustering using the Ward’s distance matrix. A constellation plot was calculated based on the dendrogram generated by the hierarchical clustering analysis. All statistical analyses were performed using JMP v14.0.

Glycoworkbench 2.0 software was used for calculation of the N-glycan structures (permethylated and perdeuterated).

### 2.7. Determination of N-Glycan Profile from Tissue Samples

Permethylated N-glycans isolated from colorectal cancer tissues (light isotopes) were joined to deuterated N-glycans isolated from normal colorectal tissues (heavy isotopes) in a 1:1 proportion, in order to independently measure the mass spectrometry signals produced by each structure isolated from both tissues of each patient. Samples corresponding to each of the patients (n = 13) were analyzed by MALDI-TOF/MS and LC-MS and the ratio between signals produced in mass spectra or TIC for each methylated glycan was calculated by using the heavy isotope signal of the same structure as reference (I-CH3/I-CD3 signals). For MALDI-TOF/MS data only ions with areas ≥5% of the base peak were considered for qualitative and quantitative analysis. Similarly, only ions with signal/noise (S/N) ratio ≥1000 were used to evaluate altered ions in the LC-MS methodology. Glycoworkbench 2.0 software and previous published N-glycans structures described in normal colorectal, colorectal cancer tissues, and serum human N-glycome were used as reference for structural identification [31,32,33,34].

### 2.8. Mass Spectrometry Acquisitions and Analysis

MALDI-TOF/MS and LC-MS/MS acquisition were performed exactly according to Barbosa et al. [30].

### 2.9. MALDI-TOF/MS

Each sample was dissolved in 30 µL of acetonitrile, mixed with 2,5-dihidroxybenzoic acid (DHB) ionization matrix (10 mg/mL in acetone containing sodium acetate 3 mM) in a proportion of 1:3, spotted on a MALDI-target plate and dried at room temperature. An UltraFlex III extreme mass spectrometer (Bruker Daltonics) operating in the positive mode and controlled by FlexControl 4.0 software was used for acquisitions. The mass range analyzed was between *m*/*z* 1500–4500. Before acquisitions, the mass spectrometer was calibrated by using Peptide Calibration Standard II (Bruker Daltonics). Mass spectra analysis as well as calculation of the areas under peaks of interest were performed using FlexAnalysis 3.4 software. Spectra were re-calibrated using the ions at *m*/*z* 1835.9, 2040.0, 2431.2, 2792.4, 2910.2, 3602.8, 4413.2, and 4597.3 as internal standards. Mass lists were generated applying Snap Peak Detection Algorithm, TopHat Baseline Subtraction, and Signal to Noise Threshold equal to 6. Values obtained were exported to Microsoft Excel for further calculations and statistical analyses.

### 2.10. ESI-LC-MS/MS

All samples were dissolved in 40 µL of MeOH containing 10 mM sodium acetate. Mass spectrometry acquisitions of each sample were performed by automatic injection of 6 µL through an ekspert^TM^ultralc 100-XL chromatography system (Eksigent, Dublin, CA, USA) equipped with a Kinetex 2.6 µm C_18_ 100 Å (50 × 2.1 mm) LC Column connected to a TripleTOF 5600+ mass spectrometer (Sciex, Concord, ON, Canada) housing a DuoSpray Ion Source. The solvents used for reversed phase chromatography were Milli-Q H_2_O containing formic acid 0.1% (solvent A) and MeOH containing formic acid 0.1% (solvent B). Samples eluted across a linear gradient of solvent B ranging from 30 to 95% in 10 min with a flow rate of 0.4 mL/min and at a constant temperature of 40 °C. Ion source operated in the positive mode at a temperature of 650.0 °C. Mass spectrometer worked in the high-resolution mode with curtain gas equal 15 and mass rage of acquisitions between *m*/*z* 800–2000. Other acquiring parameter: number of cycles = 2043; period cycle time = 525 ms; polarity = positive; pulser frequency = 13.569 kHz; and accumulation time = 500.00 ms. Mass spectrometer was calibrated using APCI positive calibration solution before acquisitions until obtaining an accuracy of ~0.2 ppm. MultiQuant^TM^ 3.0.2 software (Sciex, Concord, ON, Canada) was used for calculation of peak areas. LC-MS acquisitions were re-calibrated using peaks *m*/*z* 702.86381, 838.83862, 974.81343, 1110.78824, 1246.76305, 1382.73786 resulting from Sodium acetate aggregation as internal standards. Ions were extracted using theoretical glycan masses ±0.005 Da. Area values were exported to Excel for further calculations and statistical analyses.

Fragmentation spectra (MS/MS) of N-glycans were acquired in IDA (Information Dependent Acquisition) mode. Ions with charge state ranging from 2 to 4 were automatically selected and fragmented using dynamic collision energy mode. MSConvert (ProteoWizard 3.0) was used for conversion of LC-MS/MS data from WIFF to mzXML format. The fragmentation spectra were automatically annotated using GRITS Tollbox 1.2 software. The parameters of annotation were: 5.0 ppm of accuracy MS; 500 ppm of accuracy MSn; 5.0% of fragment intensity cut-off; perMe or perDMe derivatization type; free reducing end; N-glycans—1190 glycans search data base; maximum of 3 cleavages; maximum of 1 cross ring cleavages; glycosidic cleavages of B, Y, C, and Z series; cross ring cleavages of A and X series; maximum of 4 charges as sodium adducts. The annotated spectra were exported to Excel and printed to PDF files. Additionally, GlycoWorkbench 2.1 build 146 software was applied to analyze and annotate some spectra whose precursor ion mass matched accurately to N-glycans that were not annotated by the GRITS Toolbox.

## 3. Results

Fifteen patients previously diagnosed with sporadic colon cancer underwent surgical procedures in the Division of Colorectal Surgery at the University Hospital of Brasilia and were enrolled in the present study. Out of these, two patients were excluded: a 57-year-old patient who refused to perform a genetic test for a germinative pathologic mutation and a 37-year-old patient with a well-differentiated adenocarcinoma in transverse colon, as he tested positive for germinative pathological mutation in MLH1 gene. The clinicodemographic characteristics of the remaining 13 patients are listed in Table 1.

Blood serum and CRC tissue samples were obtained from cancer patients plus blood serum from 11 out of 15 enrolled control individuals for the qualitative and quantitative analysis of N-glycans using mass spectrometry techniques. Four individuals from the control group had mild inflammatory findings on colonoscopy and were excluded. The clinicodemographic characteristics of the 11 controls are listed in Appendix A.

No statistical difference was observed between age of colon cancer patients (mean 59.2 years-old, CI 95%, S.D. 11.9) and control group (mean 58 years-old, CI 95%, S.D, 5.8).

### 3.1. Patients with Colon Cancer Present Differential N-Glycans in Their Blood Serum in Relation to Normal Individuals

N-glycans from the blood serum of thirteen (13) colon cancer patients and eleven (11) control individuals were derivatized with iodomethane (I-CH_3_) and mixed, at a 1:1 (*v*/*v*) ratio, with N-glycans derivatized with heavy iodomethane (I-CD_3_) obtained from a pool of blood serum from control individuals. Samples were first analyzed by MALDI-TOF/MS in positive mode for a preliminary evaluation of ions in the range of *m*/*z* 1500–5000. As anticipated, MALDI-TOF/MS mass spectra of samples evidenced numerous ions compatible with N-glycans found in human blood in permethylated and perdeuterated forms, detected mostly as sodium adducts. The mass spectrum of the blood serum of one cancer patient is provided in Figure 1A for illustration, along with the proposed structures for selected ions compatible with N-glycans. To provide a more comprehensive coverage of serum N-glycans and to acquire relevant structural data, reverse phase LC-MS experiments were performed. The total ion chromatogram (TIC) of the MS analysis of the serum of a CRC patient and the extracted ion chromatograms (XICs) for the methylated and deuterated forms of the biantennary N-glycan HexNAc4Hex5NeuAc2 (Nr. 52 in Table 2), the most abundant in human serum, are shown in Figure 1B.

MS/MS spectra were acquired for ions detected in the LC-MS/MS analyses of the blood serum of all evaluated subjects. Ions were selected and fragmented according to the acquisition parameters described in the Material and Methods section and corresponding spectra were submitted to automated software interpretation. These data provide further structural information for ions detected in MS mode, confirming that they indeed correspond to permethylated and deuterated N-glycans, besides narrowing down to a restricted number of N-glycan structural isomers. As an illustration, the MS/MS spectra of HexNAc4Hex5Fuc1NeuAc2 (precursor mass [M + 3Na]^3+^ = 1004.1560 Da) and HexNAc2Hex7 (precursor mass [M + 2Na]^2+^ = 1005.4862 Da) in their permethylated forms, are provided in Figure 2A,B. Fragmentation spectra of various other N-glycan ions detected in the present study in permethylated and deuterated forms are provided as Appendix A. These, along with other human plasma N-glycans obtained from specialized databases, were compiled in a table, which is provided in full as Appendix A and in short form in the manuscript body as Table 2. Table 2 lists selected N-glycan compositions and their proposed structures, the charge state, and molecular masses used for the development of quantification methods, as well as the availability of structural information from MS/MS spectra. It also introduces a glycan numbering system that will be adopted throughout the present manuscript. Given that multiple ions did not produce reliable MS/MS spectra due to low signal intensity, only ions for which tandem MS spectra were obtained either for the permethylated or perdeuterated forms were considered in the development of quantitation methods.

Quantification methods for N-glycans in the serum of CRC and control patients were developed for MALDI-TOF/MS and LC-MS/MS data. These methods are based on the calculation of area ratios between ions from N-glycans derivatized with regular versus deuterated iodomethane (-CH_3_/-CD_3_ ratios) using a “nearest neighbor” approach, similarly to a recent publication [30]. After the calculation of area ratios for all N-glycan pairs for each patient, data was further normalized using HexNAc4Hex5NeuAc2 (Nr. 52) as reference. The quantification method developed for MALDI-TOF/MS data included twenty-nine (29) N-glycans of various compositions (Appendix A) and only one of them had significantly different area ratios in CRC patients in relation to controls, as shown in Figure 3A. The N-glycan HexNAc5Hex5NeuAc1 (Nr. 49) was upregulated in CRC patients, according to the Mann–Whitney U test with α = 0.01, as detailed in Table 3.

A similar relative quantification approach was developed for the LC-MS/MS data; however, it included fifty-three (53) N-glycans of varying compositions evaluated herein (Appendix A). Those that were significantly altered in CRC patients in relation to control individuals are reported in Table 3 and in Figure 3B. Twenty-six (26) N-glycans of varying composition were found to be significantly altered in the plasma of cancer patients. Among these, HexNAc5Hex4 (Nr. 22) and HexNAc2Hex7 (Nr. 17) were the ones with the highest statistical significance in the Mann–Whitney U test. Three N-glycans were found at a higher relative concentration in the serum of cancer patients: the mannose-rich HexNAc2Hex7 (Nr. 17), the fucosylated bi-antennary glycan HexNAc4Hex5Fuc1NeuAc2 (Nr. 58), and the tetra-antennary HexNAc6Hex7NeuAc3 (Nr. 79). 

To investigate whether the alterations in the relative abundance of serum N-glycans of CRC patients detected in LC-MS/MS analyses can be used as a classification tool, normalized -CH_3_/-CD_3_ area ratios of subjects (n = 24, 13 CRC patients, 11 controls) were submitted to a hierarchical clustering algorithm using the Ward’s method. According to the constellation plot provided in Figure 4, calculated from resulting dendrograms, two main clusters were readily apparent: cluster one has twelve (12) members, and it holds ten out of the eleven (10/11) control individuals as well as two out of thirteen (2/13) cancer patients. Cluster two also has twelve (12) members and is formed by eleven out of thirteen (11/13) colon cancer patients and one (1) control individual.

### 3.2. N-Glycan Profiles in CRC Tissue Are Heterogeneous, However, Some Glycan Compositions Are Consistently Altered in Patients

Colonic tumor samples were obtained and their N-glycans were extracted and quantified in relation to adjacent tissue of the same patients. A similar quantification methodology was developed, however, this time, N-glycans from tumor tissue were derivatized with I-CH_3_, while those from normal tissue were derivatized with I-CD_3_. This implies that the area ratios of evaluated N-glycans show whether these are relatively increased (ratio > 1) or decreased (ratio < 1) in tumor tissues in relation to normal tissue samples. Mass spectrometric analyses of N-glycans in tissue extracts indicate that the CRC tumor microenvironment is highly heterogeneous. The MALDI-TOF/MS mass spectra of patients were highly variable in terms of the quality and quantity of N-glycan ions (Appendix A). Thus, given such high variance, only ions with areas ≥5% of the base peak were considered for the development of quantitative methods for MALDI-TOF/MS data. Similarly, only ions with S/N ratio ≥1000 were considered for quantitation in the LC-MS/MS methodology. To evaluate N-glycans that were consistently altered in samples, a binomial test was applied and those compositions presenting statistical significance are listed in Table 4. The (-CH_3_/-CD_3_) ratios were calculated and were used to build box plots (Figure 5). 

Thirteen (13) ions matching N-glycans were consistently detected in the MALDI-TOF/MS spectra of the tissue extract. Out of these, two (2) were increased in relation to normal tissues: the oligomannosidic structures HexNac2Hex5 (Nr. 6) and HexNac2Hex6 (Nr. 10) (Figure 5). A full list of all detected ions and their area ratios is available as Appendix A.

Twenty-four (24) N-glycans presented consistent ion signals in LC-MS/MS analyses of tissue samples and were subjected to further evaluation (Appendix A). One N-glycan composition was decreased in cancer tissues, HexNAc2Hex7 (Nr. 17), with median -CH_3_/-CD_3_ ratio equivalent to 2.0 × 10^−2^. Ten other N-glycan compositions were increased. Out of these, one was an oligomannosidic structure, HexNAc2Hex9 (Nr. 33), with a median -CH_3_/-CD_3_ ratio of ~49 units, being thus highly discriminative in the CRC tumor tissues evaluated herein. (Figure 5). Still other hypogalactosylated and hyposialylated complex glycans were more abundant in tumoral tissue, like N-glycans HexNAc5Hex4 (Nr. 22), HexNAc4Hex4NeuAc1 (Nr. 26), HexNAc5Hex4NeuAc1 (Nr. 36), and HexNAc5Hex5Fuc1NeuAc2 (Nr. 64). Complex fucosylated tri- and tetra-antennary glycans presenting terminal galactose residues were also up-regulated, like HexNAc6Hex6Fuc1 (Nr. 57) and HexNAc6Hex7Fuc2 (Nr. 67).

### 3.3. Correlation between Relative Levels of N-Glycans in Serum and Tissue of CRC Patients

Eight N-glycan ions presented significantly altered area ratios in both the serum and tissue of colon cancer patients in relation to controls, all of them detected in LC-MS/MS analyses of their respective biological materials. None of these were consistently increased or decreased in tissue and serum. For example, while the oligomannosidic glycan HexNAc2Hex7 (Nr. 17) was increased in the serum of cancer patients, it was decreased in tissue. Other N-glycans, like HexNAc5Hex4 (Nr. 22), HexNAc4Hex4NeuAc1 (Nr. 26), and HexNAc5Hex9 (Nr. 33), while significantly decreased in serum, were increased in the CRC tissue samples of the same patients. Quantitative correlations between altered glycans in tissue and their corresponding levels in serum were sought, and not a single composition showed positive or negative correlation. These analyses jointly illustrate the complex relationship between N-glycan compositions in tumoral tissue and in the serum of CRC patients.

## 4. Discussion

The present work reports the application of a mass spectrometry-based quantitative methodology for the evaluation of altered N-glycans in the blood serum and cancerous tissue of CRC patients. By quantifying N-glycans in healthy individuals in relation to a control pool, it is possible to estimate the variance for each glycan composition found in the normal population and therefore quantify putative alterations in the serum of CRC patients. Similar methodology was applied to identify diagnostically relevant N-glycans present in the plasma of Type II Congenital Disorder of Glycosylation patients, and as result, it was observed significant alterations even in glycans of low abundance [30].

Several N-glycans were found to be quantitively altered in the blood serum of CRC patients in relation to control individuals. Initially, fast MALDI-TOF/MS evaluations identified one up-regulated molecule, while LC-MS/MS analyses identified 23 down-regulated and 3 up-regulated N-glycan compositions in CRC patients. This suggests that while MALDI-TOF/MS might be useful as a preliminary tool to assess the quality of the material, ion suppression, lack of mass accuracy, and lower resolution power might limit its capacity to fully determine significant alterations in serum N-glycans of CRC patients, in consonance with previous reports [30]. Overall, the predominant alteration identified in LC-MS/MS analyses of the serum of CRC patients was down-regulation of N-glycans from a large variety of structural families. These include mannose-rich structures, such as HexNAc2Hex8 (Nr 25), hybrid-type structures, such as HexNAc3Hex5Fuc1NeuAc1 (Nr. 31) and several complex-type structures. Among the latter, down-regulation was found for hypogalactosilated and hyposilialated N-glycans, such as HexNAc5Hex4 (Nr. 22) and HexNAc4Hex5Fuc1NeuAc1 (Nr. 44), as well as highly branched structures, such as HexNAc6Hex6 (Nr. 51) and HexNAc5Hex6NeuAc1 (Nr. 55). Up-regulations were more specific, and three N-glycans were found to be significantly increased in CRC: the high-mannose HexNAc2Hex7 (Nr. 17), the complex-type core-fucosylated bi-antennary HexNAc4Hex5Fuc1NeuAc2 (Nr. 58), and the highly branched HexNAc6Hex7NeuAc3 (Nr. 79). Based on previous work, which describe the 24 glycoproteins that contribute most with the plasma N-glycome, HexNAc2Hex7 (Nr. 17), or Man7 according to the Oxford nomenclature, is found in both Immunoglobulins E and M [24]. Alterations in the levels of serum immunoglobulins, mainly IgA, IgG, and IgM, have been consistently reported in CRC, as well as other cancer types [35]. HexNAc4Hex5Fuc1NeuAc2 (Nr. 58), or FA2G2S2, is found on eleven different glycoproteins, and serotransferrin, alpha-1-antitrypsin, alpha-2-macroglobulin, IgA1, and IgGs are the ones presenting highest serum concentrations. Nonetheless, core-fucosylated compositions, catalyzed by enzyme fucosyltransferase-VIII (FUT-8), have been associated to N-glycosylation of IgGs since hepatic tissues hardly express FUT-8 activity [36]. Therefore, the observed increase of core-fucosylated biantennary di-sialo N-glycans may reflect the systemic response of B cells in producing a special subtype of N-glycosylated IgG against the extraneous tumor tissue [37,38].

N-glycan HexNAc6Hex7NeuAc3 (Nr. 79), or A4G4S3, is found mostly on Alpha-1-acid glycoprotein (AGP), an acute phase protein associated with the negative modulation of the complement system and transport of lipophilic compounds [39,40]. The fact that few N-glycan structures were found to be augmented in CRC deserves further attention, as it might result from either microheterogeneity in the fore mentioned plasma glycoproteins or an augmented expression of novel proteins preferentially decorated with these structures.

The alterations in the serum of CRC patients described herein coincide partially with the literature. A previous work using HILIC chromatography with fluorescence detection reported a general decrease in bi-antennary core fucosylated structures containing mono or di-galactosylated moieties involving at most one sialic acid, while a general increase was observed in glycans from highly branched structures (>3 GlcNAc antennae), rich in galactose (>3 galactose) and in sialic acid (>3 sialic acids) [27]. In our study, multiple mono and di-galactosylated structures were indeed decreased, however, the only over-abundant high-branched structure was HexNAc6Hex7NeuAc3 (Nr. 79). In another study, in which a PLS-DA model was applied to MALDI-TOF/MS mass spectra of the total serum N-glycome of controls vs. cases, several N-glycans were found to be differential in CRC patients, some down- and others up-regulated. Although some alterations agree with our results, glycan H5N4F1L2, composition HexNAc4Hex5Fuc1NeuAc2 (Nr. 58 in Table 2), was downregulated in this study, contrasting with our findings [41]. This indicates that the methodology used for data acquisition is a sensitive feature in the measurement of quantitative alterations in N-glycans of CRC patients, which might be difficult inter-study comparisons. A summary of our findings compared to the actual literature on N-glycosylation in serum of CRC patients were compiled in Appendix A, provided in Appendix A. 

Therefore, one feature worthy of evaluation is whether the alterations described herein can, collectively, be used to discriminate CRC patients from controls in an unbiased larger sample population.

In the present work, a hierarchical clustering algorithm was applied to the -CH_3_/-CD_3_ ratios calculated from LC-MS/MS experiments for the 53 N-glycans quantified in the blood serum of CRC patients and controls. This is an unsupervised multivariate statistical method and no information regarding group memberships were provided *a priori* [42]. Following the clustering of data, approximately eighty-five (85%) of the CRC patients were grouped in a single cluster, while nearly ninety-one (91%) of the control individuals were grouped in the alternative cluster. This suggests that the present methodology holds potential in the diagnosis of CRC patients and that more sophisticated statistical models might result in even better discrimination. However, it is our understanding that sample size is a limitation in our study and that it should be significantly increased prior to the development of more sophisticated models.

It has been suggested that the glycans encountered in human serum are pooled from all cells in the body, and that these might be altered by pathologies such as cancer [13,17]. N-glycans in CRC and normal colonic tissues from the same patient were evaluated and their putative correlations with serum N-glycans were explored. Heterogeneity was a hallmark in our findings, indicating that the tumoral micro-environment may vary significantly in N-glycan content among individuals. This may be attributed to the very heterogeneity observed in our samples, comprised by stage II and III patients with a variable RAS/RAF mutational profile, not rarely found in sporadic CRC. In fact, stage II and III make up to 70% of all CRC cases, whilst mutation in RAS/RAF oncogenes may respond to 40% of CRC as well [43,44]. Albeit highly heterogenous, not a single composition was found consistently over- or under-expressed when tumor tissue samples were stratified by primary CRC location, i.e., right- or left-sided, stage, or mutational status (data not shown). These findings are in line with the Holm et al. study, in which no difference was found in non-acidic N-glycans by stage or primary location, even though a N-glycan signature in tumor stroma has been attributed to stage II CRC samples [19,21].

Nevertheless, our analysis focused in uncovering those glycans that were consistently altered in CRC patients, and the main findings concern an overall increase in oligomannosidic, bi-antennary mono-galactosylated, and highly branched structures, some of them presenting core fucosylation. Again, these results overlap only partially with the literature, as individual structures previously found to be up- or down-regulated in CRC tissue were not coincident [15,19]. Oligomannosidic and mannose-rich N-glycans are thought to be abundant in cancerous tissues due to reduced expression of the enzyme MGAT1 [15,45], responsible for the transformation of high-mannose to hybrid type glycans, while fucosylated structures, mostly core-fucosylated, are thought to arise from an increase in the expression of the enzyme FUT8 [18,36]. This glycosyltransferase catalyzes the addition of the α1-6-linked fucose to the core GlcNAc and has been consistently associated with several types of cancer and to poor prognosis in CRC [18].

Similarly, increase of branched structures has been associated with malignant transformation in many tumors, as in CRC [46]. It is believed that in tumor cells, *Ras* oncogene mutation leads to overexpression of MGAT-5, the glycosyltransferase responsible for adding a β-1,6-N-acetylglucosamine to the core of N-glycans [47]. In turn, branching compositions on cell surface promotes invasion and decrease cell-cell adhesion mediated by E-cadherin, both factors associated with worse prognosis in CRC [48].

Bi-antennary mono-galactosylated and mono-galactosylated caped with one sialic acid compositions were overrepresented in tumor samples. Curiously, these forms are not usually found increased in CRC [19]. On the contrary, MS-based studies have shown a decrease rather than increase of bi-antennary compositions, mostly bisected forms [15,16]. Similar with observed in serum, bi-antennae N-glycans are commonly attached to IgG. Therefore, in tumor tissues, predominance of the aforementioned special type of complex bi-antennary compositions may indicated a shift of micro-heterogeneity in glycosylation sites of immunoglobulins G, in favor to a pro-inflammatory subgroup of glycoforms, more effective in the host response to tumor tissue. A summary of our findings compared to the actual literature on N-glycosylation in CRC tissue were compiled in Appendix A, provided in Appendix A.

However, the N-glycans profile of tumor samples hardly mirrored the N-glycans found in serum samples of CRC patients. The lack of agreement between qualitative N-glycans profiles is not surprising, as these profiles might arise from distinct sets of proteins, as previously suggested for ovarian and colorectal cancer [11,49]. In this setting, serum glycosylation is likely to reflect the N-glycosylation of acute phase response, like in IgG and α1-acid glycoprotein, rather than glycan composition shed by tumor tissues into blood [50,51].

Our study focused mainly on compositional nature of glycans. Relative quantification by MS analysis, of released and labeled N-glycan compositions from biologic samples poses, in itself, a real challenge in microanalytic research, since biomarker discovery relies on highly reproductible methods [10,31,34]. Albeit the number of compositions found in our MALDI-TOF/MS analysis is in line with previous MALDI-TOF/MS-based studies [15,41,52], lack of agreement between MALDI-TOF/MS and LC-MS/MS measurements is noteworthy. In contrast to LC-MS/MS analysis, MALDI-TOF/MS analysis yielded only three compositions capable of discriminating CRC patients from healthy controls, in both serum and tissue samples. In our understanding, ion signal suppression is responsible for this discrepancy. In this sense, we observed that some compositions may have not reached statistical significance merely due to its low concentration in serum or in tissue samples, the ones more likely to suffer highest influence from ion signal suppression. Thus, our preliminary set of compositions relied on MALDI ionization acquisitions whilst our quantitation of structures were restricted to LC-MS/MS analysis.

Moreover, most of the current knowledge of the N-glycan profile in cancer has been based on quantification methods by LC-MS technology, such as adopted in this study, with characterization of N-glycan in a compositional level by accurate mass profiling [10,14,15,16,17,19,20,21,30,31,41]. Notwithstanding, we are aware that in-depth analysis of linkage positions and retention times of isomers or differential anomeric forms in LC/MS acquisitions, may add new insights on glycosylation as a source of biomarkers in cancer, what will surely deserve close attention in further studies.

## 5. Conclusions

In conclusion, our study revealed a global decrease of galactosylated compositions present in serum of CRC patients by LC-MS technology. Conversely, three compositions were upregulated in serum pointing to a subset of structures highly discriminative between cases from healthy controls. Moreover, our results suggest this panel holds potential in clinical setting with applications in large scale by automated MS analysis. Albeit heterogeneous, a commonality of high mannose and branched compositions could be identified upregulated in tumor tissues. However, some new compositions found an increase in tumor, as oligomannosidic and bi-antennary mono-galactosylated glycans, have hardly been described in CRC, what will surely deserve closer attention in further studies. Curiously, no single N-glycan increase in tissues was found upregulated in serum, which poses the question where serum compositions in CRC patients are from.

## Figures and Tables

**Figure 1 biology-10-00343-f001:**
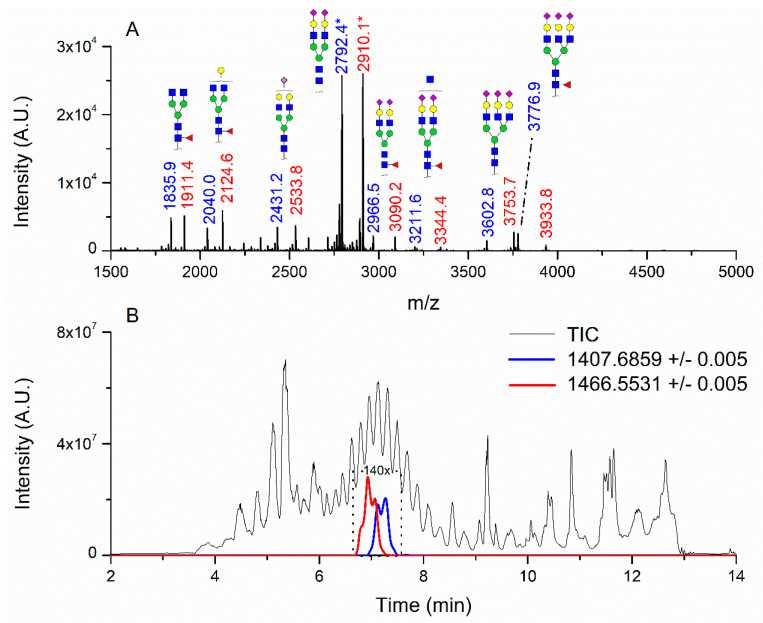
Analysis of serum N-glycans by mass spectrometry. (**A**) MALDI-TOF/MS spectrum of blood serum N-glycans of patient 7 mixed with those from the control pool in positive mode in the range *m*/*z* 1500–5000. Ions whose masses correspond to methylated N-glycans are depicted in blue, while those from deuterated structures are given in red. The two most intense peaks correspond to the N-glycan HexNAc4Hex5NeuAc2 in its permethylated form ([M + Na]^+^ = 2792.4 Da), obtained from patient 7, and deuterated forms ([M + Na]^+^ = 2910.1 Da), obtained from control pool. Proposed structures were drawn for their corresponding N-glycan compositions and are depicted on top of their corresponding ions. (**B**) LC-MS analysis showing total ion chromatogram (black line) and extracted ion chromatograms for the N-glycan HexNAc4Hex5NeuAc2 in the permethylated form ([M + Na]^2+^ = 1407.6859 Da) isolated from patient 7 and deuterated form ([M + Na]^2+^ = 1466.5531 Da) obtained from control pool. Masses and XICs in blue and red indicate light (methylated) and heavy (deuterated) derivatized N-glycans, respectively. The asterisks drawn in inset A correspond to the same N-glycans analyzed in inset B, although the latter in a different charge state than the former.

**Figure 2 biology-10-00343-f002:**
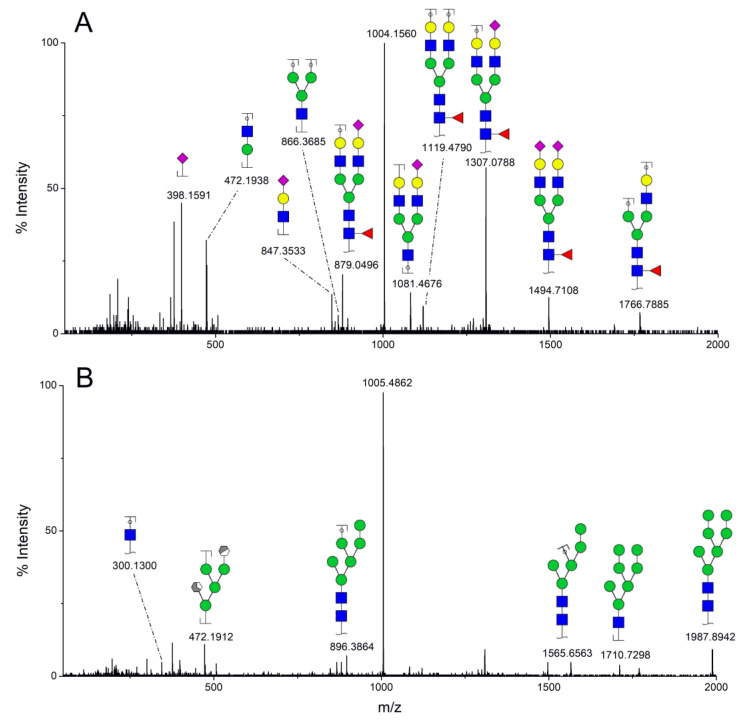
MS/MS spectra of permethylated N-glycans obtained in a reverse-phase LC-MS/MS analysis of a colon cancer patient, (**A**) HexNAc4Hex5Fuc1NeuAc2 (precursor mass [M + 3Na]^3+^ = 1004.1560 Da, Nr. 58 in Table 2) and (**B**) HexNAc2Hex7 (precursor mass [M + 2Na]^2+^ = 1005.4862 Da, Nr. 17 in Table 2). Data interpretation was performed automatically by the software GRITS.

**Figure 3 biology-10-00343-f003:**
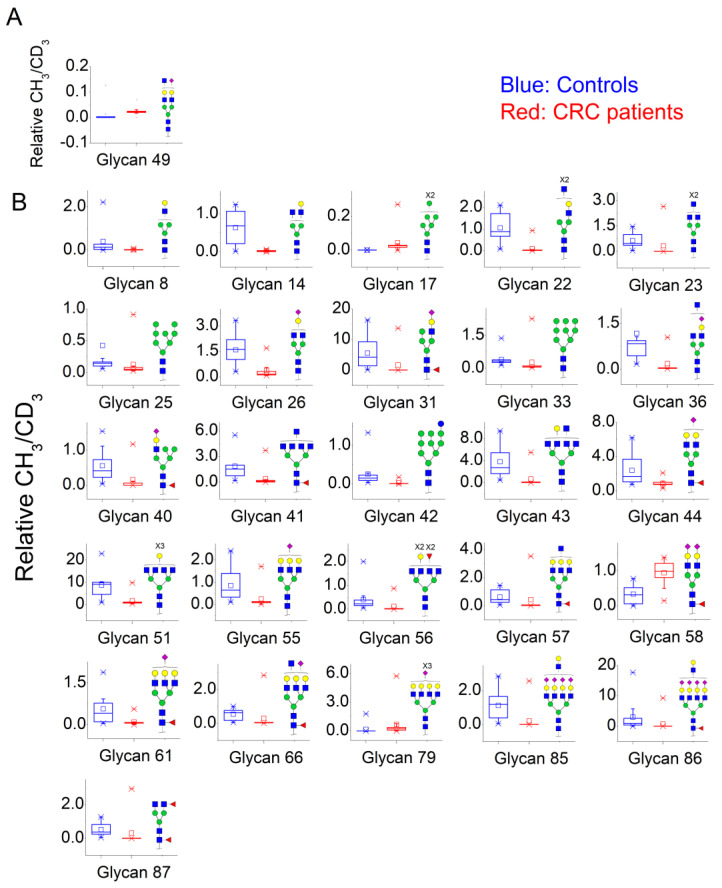
Box plots showing relative abundance of up- and down-regulated N-glycans isolated from blood serum of cancer patients and control patients (Mann–Whitney U test, α = 0.01) analyzed by (**A**) MALDI-TOF/MS and (**B**) LC-MS/MS. The lower and upper whiskers represent the 5th and 95th percentile, respectively, while the median is represented as a straight line, the mean by an open box (⎕), and data outliers are represented by the X symbol.

**Figure 4 biology-10-00343-f004:**
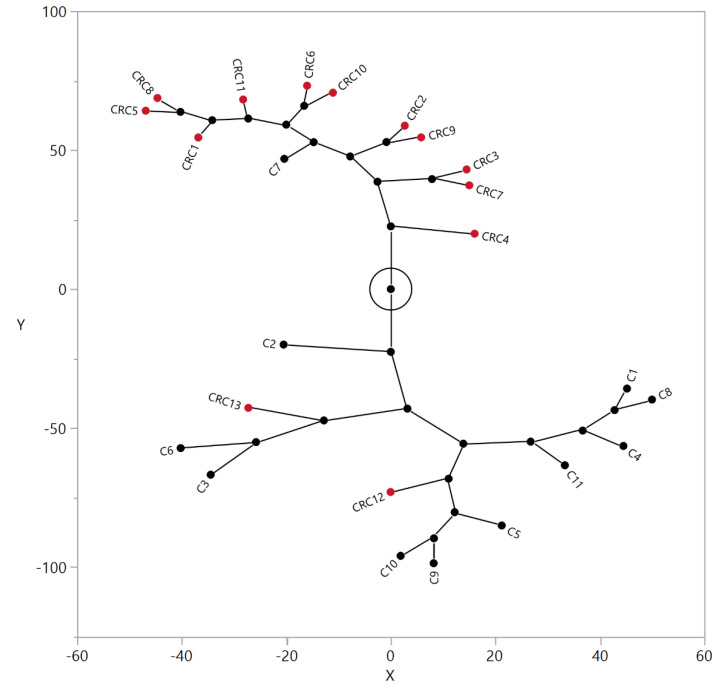
Constellation plot generated for the N-glycans in the blood serum of CRC patients and control individuals. The normalized and standardized -CH_3_/-CD_3_ ratios for fifty-three (53) N-glycans obtained by LC-MS/MS analyses of the blood serum of CRC patients (●) and control individuals (●) were submitted to a hierarchical clustering algorithm using the Ward’s method and the resulting dendrogram was depicted as a constellation plot. X and Y axes are dimensionless.

**Figure 5 biology-10-00343-f005:**
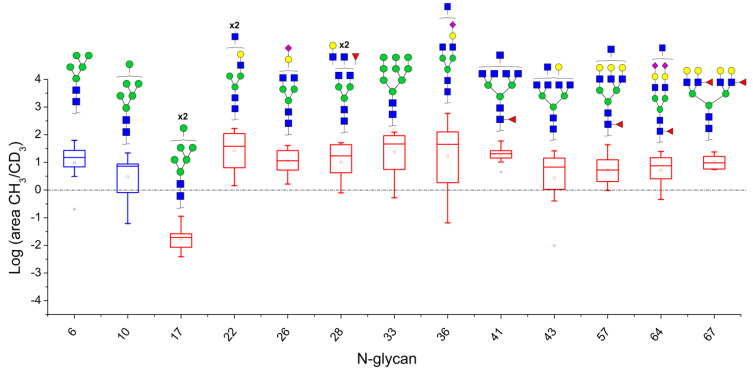
Box plots showing up- and down-regulated N-glycans isolated from tumor and normal colonic tissues of colon cancer patients analyzed by MALDI-TOF/MS (blue boxes) and LC-MS/MS (red boxes). The lower and upper whiskers represent the 5th and 95th percentile, respectively, while the median is represented as a straight line, the mean by an open box (⎕), and data outliers are represented by the X symbol.

**Table 1 biology-10-00343-t001:** Clinicodemographic characteristics of patients.

Patient	Genre	Age (years)	Location	Stage	CEA (ng/mL)	Tumor Diameter (cm)	Mutational Nras/Kras/Braf Status
1	M	37	TRANSVERSE	II	1.48	3.70	Kras mutated
2	F	36	SIGMOID	II	1.65	2.80	No
3	M	65	CECUM	II	1.60	4.30	Kras mutated
4	M	55	DESCENDING	III	5.30	2.60	No
5	M	60	SIGMOID	II	7.70	4.80	No
6	F	56	SIGMOID	III	2.10	3.70	No
7	M	67	ASCENDING	III	2.30	6.00	Braf mutated
8	F	64	CECUM	III	11.25	4.10	No
9	F	63	CECUM	II	6.51	6.20	No
10	F	54	SIGMOID	III	3.30	3.50	Braf mutated
11	F	74	SIGMOID	III	6.75	1.20	No
12	M	64	CECUM	III	5.20	7.20	Kras mutated
13	F	75	SIGMOID	II	1.35	4.60	Kras mutated

CEA—carcinoembryonic antigen.

**Table 2 biology-10-00343-t002:** N-glycan number, structure, accurate mass, mass used for quantification, corresponding adduct, and the heavy reference number used for LC-MS/MS and MALDI-TOF/MS quantification methods.

Nr.	Proposed Structure, Theoretical Mass, and Composition	Mass Used for Quantitation	Adduct	Heavy Reference		Structure Confirmed by MS/MS Data ^a^
				LC-MS/MS	MALDI	
6	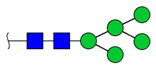 HexNac2Hex5	1579.7826	[M + Na]^+^	52	6	✓
8	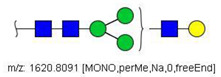 HexNAc3Hex4	1620.8090	[M + Na]^+^	19	-	*
10	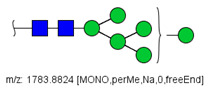 HexNAc2Hex6	903.4358	[M + 2Na]^2+^	52	6	✓✓
14	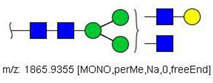 HexNAc4Hex4	1865.9355	[M + Na]^+^	19	19	✓✓
17	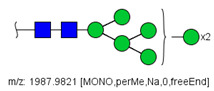 HexNAc2Hex7	1005.4857 *	[M + 2Na]^2+^	52	6	✓✓
22	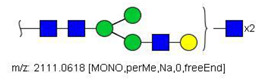 HexNAc5Hex4	1067.0255	[M + 2Na]^2+^	19	-	*
23	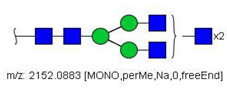 HexNAc6Hex3	1087.5388 *	[M + 2Na]^2+^	13	-	✓
25	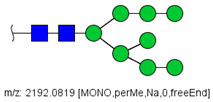 HexNAc2Hex8	1107.5356	[M + 2Na]^2+^	52	6	✓✓
26	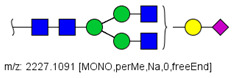 HexNAc4Hex4NeuAc1	1125.0492	[M + 2Na]^2+^	26	35	✓✓
28	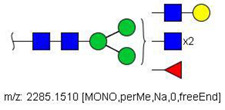 HexNAc5Hex4Fuc1	1154.0701	[M + 2Na]^2+^	19	19	✓
31	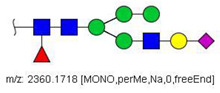 HexNAc3Hex5Fuc1NeuAc1	1191.5805	[M + 2Na]^2+^	27	-	✓✓
33	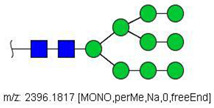 HexNAc2Hex9	1209.5855	[M + 2Na]^2+^	52	6	✓✓
36	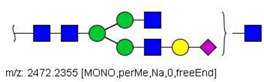 HexNAc5Hex4NeuAc1	1247.6123	[M + 2Na]^2+^	52	-	✓✓
40	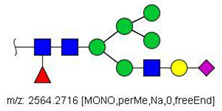 HexNAc3Hex6Fuc1NeuAc1	1293.6304 *	[M + 2Na]^2+^	27	-	✓✓
41	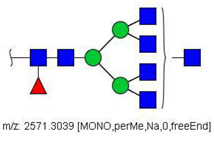 HexNAc7Hex3Fuc1	1297.1465	[M + 2Na]^2+^	75	-	*
42	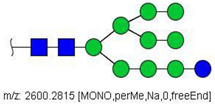 HexNAc2Hex10	1311.6353	[M + 2Na]^2+^	52	-	✓
43	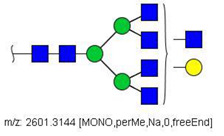 HexNAc7Hex4	1312.1518	[M + 2Na]^2+^	72	-	✓
44	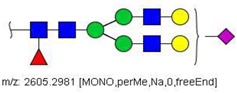 HexNAc4Hex5Fuc1NeuAc1	1314.1437	[M + 2Na]^2+^	44	58	✓✓
49	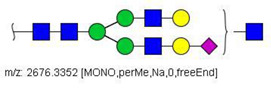 HexNAc5Hex5NeuAc1	907.4379	[M + 3Na]^3+^	35		✓
51	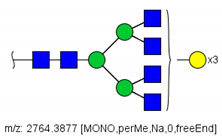 HexNAc6Hex6	936.7887	[M + 3Na]^3+^	52	35	✓
52	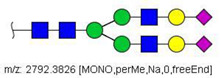 HexNAc4Hex5NeuAc2	1407.6859	[M + 2Na]^2+^	52	52	✓✓
55	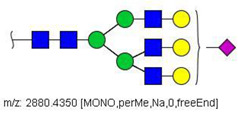 HexNAc5Hex6NeuAc1	975.4711	[M + 3Na]^3+^	72	35	✓✓
56	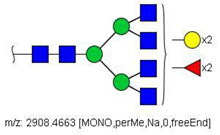 HexNAc6Hex5Fuc2	984.8149	[M + 3Na]^3+^	27	-	✓
57	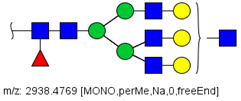 HexNAc6Hex6Fuc1	994.8184	[M + 3Na]^3+^	27	-	✓
58	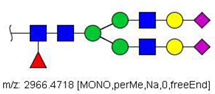 HexNAc4Hex5Fuc1NeuAc2	1004.1501	[M + 3Na]^3+^	58	58	✓✓
61	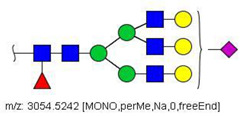 HexNAc5Hex6Fuc1NeuAc1	1033.5009 *	[M + 3Na]^3+^	27	-	✓
64	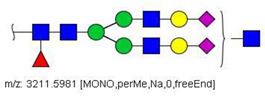 HexNAc5Hex5Fuc1NeuAc2	1617.2937	[M + 2Na]^2+^	27	64	✓
66	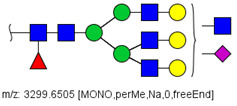 HexNAc6Hex6Fuc1NeuAc1	1115.2096	[M + 3Na]^3+^	27	-	✓
67	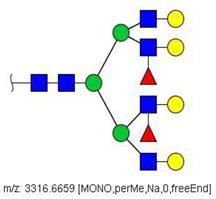 HexNAc6Hex7Fuc2	1669.8275 *	[M + 2Na]^2+^	27	-	✓
79	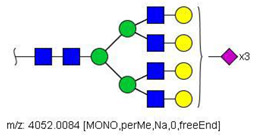 HexNAc6Hex7NeuAc3	1030.2440	[M + 4Na]^4+^	80	52	✓
85	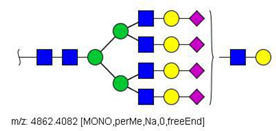 HexNAc7Hex8NeuAc4	1232.8439 *	[M + 4Na]^4+^	80	-	*
86	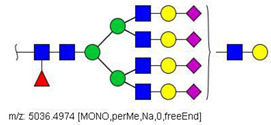 HexNAc7Hex8Fuc1NeuAc4	1276.3663 *	[M + 4Na]^4+^	80	-	✓
87	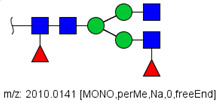 HexNAc4Hex3Fuc2	1016.5017	[M + 2Na]^2+^	13	-	✓✓

^a^ one check indicates that permethylated structure was confirmed, double check indicates that both structures (permethylated and perdeuterated) were confirmed, while asterisk indicates that only perdeuterated structure was confirmed. Legend: 

 N-acetyl glucosamine (GlcNac), ● Mannose (Man), ● Galactose (Gal), ◄ Fucose (Fuc), 
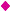
 Sialic acid (NeuAc).

**Table 3 biology-10-00343-t003:** Mann–Whitney U tests of the relative concentration of N-glycans in serum of cancer patients and controls obtained by MALDI-TOF/MS and LC-MS/MS.

Nr	N-Glycan Composition	Median Controls (n = 11)	Median Cancer Patients (n = 13)	Alteration in Cancer Patients	Z	Wilcoxon, 2-Sample Test, Normal Approximation, Prob>|Z|
MALDI-TOF/MS data						
49	HexNAc5Hex5NeuAc1	0	0.022	↑	−3.09	0.0020
LC-MS/MS data						
8	HexNAc3Hex4	0.140	0.012	↓	3.13	0.0018
14	HexNAc4Hex4	0.685	0.023	↓	3.03	0.001
17	HexNAc2Hex7	0.000	0.026	↑	−3.53	0.0004
22	HexNAc5Hex4	0.882	0.020	↓	3.77	0.0002
23	HexNAc6Hex3	0.466	0.005	↓	2.72	0.0065
25	HexNAc2Hex8	0.149	0.051	↓	2.72	0.0065
26	HexNAc4Hex4NeuAc1	1.619	0.160	↓	3.30	0.001
31	HeXNAc3Hex5Fuc1NeuAc1	4.228	0.021	↓	3.07	0.0021
33	HexNAc2Hex9	0.334	0.072	↓	3.01	0.0026
36	HexNAc5Hex4NeuAc1	0.838	0.035	↓	3.13	0.0018
40	HexNAc3Hex6Fuc1NeuAc1	0.413	0.000	↓	2.97	0.003
41	HexNAc7Hex3Fuc1	1.533	0.092	↓	3.25	0.0012
42	HexNAc2Hex10	0.147	0.008	↓	3.42	0.0006
43	HexNAc7Hex4	2.705	0.025	↓	3.36	0.0008
44	HexNAc4Hex5Fuc1NeuAc1	1.659	0.772	↓	3.01	0.0026
51	HexNAc6Hex6	9.144	0.842	↓	3.42	0.0006
55	HexNAc5Hex6NeuAc1	0.650	0.123	↓	2.66	0.0077
56	HexNAc6Hex5Fuc2	0.221	0.004	↓	3.02	0.0026
57	HexNAc6Hex6Fuc1	0.439	0.030	↓	2.66	0.0077
58	HexNAc4Hex5Fuc1NeuAc2	0.315	0.987	↑	−3.36	0.0008
61	HexNAc5Hex6Fuc1NeuAc1	0.414	0.069	↓	2.84	0.0045
66	HexNAc6Hex6Fuc1NeuAc1	0.633	0.050	↓	3.07	0.0021
79	HexNAc6Hex7NeuAc3	0.001	0.226	↑	−3.19	0.0014
85	HexNAc7Hex8NeuAc4	1.188	0.000	↓	3.44	0.0006
86	HexNAc7Hex8Fuc1NeuAc4	0.920	0.019	↓	2.95	0.0031
87	HexNAc4Hex3Fuc2	0.379	0.006	↓	2.95	0.0031

**Table 4 biology-10-00343-t004:** Relative concentration of N-glycans in samples of tumor and normal tissues from all cancer patients obtained by MALDI-TOF/MS and LC-MS/MS.

		Patient Nr.															
Nr.	N-Glycan Composition	1	2	3	4	5	6	7	8	9	10	11	12	13	Median	n/Total *	Binomial Probability
		**MALDI-TOF/MS data (area ratio)**															
6	HexNac2Hex5	29.41	62.5	-	6.88	15.33	-	7.48	20.8	0.2	3.09	14.63	-	27.09	14.98	9/10	0.0010
10	HexNac2Hex6	42.63	7.25	-	8.71	-	-	21.94	4	0.06	-	7.48	-	23.4	8.10	7/8	0.0020
		**LC-MS/MS data (area ratio)**															
17	HexNAc2Hex7	-	-	-	0.02	0.03	0.00	0.03	0.02	0.01	0.11	0.01	-	-	0.02	8/8	0.0000
22	HexNAc5Hex4	167.48	53.17	31.47	38.23	84.42	-	7.42	110.19	1.44	4.61	6.39	-	136.41	38.23	11/11	0.0000
26	HexNAc4Hex4NeuAc1	40.70	31.47	11.41	10.34	5.26	-	1.64	24.58	-	-	3.43	-	26.69	11.41	9/9	0.0000
28	HexNAc5Hex4Fuc1	51.49	17.16	46.90	-	4.20	-	-	43.48	0.79	1.84	5.27	-	19.55	17.16	8/9	0.0020
33	HexNAc2Hex9	92.23	116.52	86.94	65.26	5.54	-	4.45	32.67	0.52	-	16.76	-	123.43	48.96	9/10	0.0010
36	HexNAc5Hex4NeuAc1	140.70	113.15	59.98	142.75	17.23	0.06	1.98	35.80	0.78	1.70	56.29	-	587.41	46.05	10/12	0.0032
41	HexNAc7Hex3Fuc1	20.10	26.13	21.01	10.27	27.09	-	-	20.10	-	-	4.52	-	59.21	20.55	8/8	0.0000
43	HexNAc7Hex4	26.12	10.94	25.56	6.74	-	0.01	1.34	7.67	0.40	1.06	1.91	-	14.29	6.74	9/11	0.0059
57	HexNAc6Hex6Fuc1	-	12.13	43.48	12.63	6.20	2.52	1.76	7.29	0.97	2.32	4.52	1.76	26.01	5.36	11/12	0.0002
64	HexNAc5Hex5Fuc1NeuAc2	7.49	24.86	6.61	14.67	-	0.62	-	-	0.45	2.54	11.36	-	19.28	7.49	7/9	0.0195
67	HexNAc6Hex7Fuc2	9.62	7.77	5.74	23.82	16.50	-	-	13.98	-	-	-	-	5.50	9.62	7/7	0.0000

* n/total: n increased samples/total of samples.

## Data Availability

The datasets generated during and/or analyzed during the current study are available from the corresponding author on reasonable request. All data generated or analyzed during this study are included in this published article (and its Supplementary Information files).

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
