# Peer review of "Identification of Differential N-Glycan Compositions in the Serum and Tissue of Colon Cancer Patients by Mass Spectrometry"

_biology, 2021, doi:10.3390/biology10040343_

Round 1
Reviewer 1 Report
Coura et al have profiled the serum and tissue N-glycome of CRC patients and control donors, using two mass spectrometry-based approaches: MALDI-TOF/MS and ESI-LC/MS/MS. The authors described a serum glycosignature able to cluster separately CRC from healthy samples with 85% accuracy. Although valuable and relevant clinical cohort have been explored, this study lacks novelty and detailed comparison with glycome profile of serum and tissue of CRC previously reported in the literature. There are other major and minor several issues in the introduction, references, description of methods and data presentation that need to be addressed before the manuscript can be recommended for publication.
Major:
Introduction:
- It is not clear what is the aim of the study. By starting the introduction describing the problem of low survival and its association with the cancer staging, the authors give the impression that a biomarker for CRC prognosis will be addressed in the study, which was not the case. Please provide a background description of the current gold standard diagnostic and prognostic methods in CRC used in clinics and what is the gap that the authors aim to fill.
- The authors missed some relevant studies that profile the changes in the N-glycome of CRC tissues (e.g Sethi et al, 2015, Glycobiology, https://doi.org/10.1093/glycob/cwv042, Kaprio et al., Mol Cell Proteomics, DOI:https://doi.org/10.1074/mcp.M114.041632, Boyaval, et al., 2021, Mol Cell Proteomics, DOI: 10.1074/mcp.RA120.002215).
- The authors need to state what is the main difference between the current study and the previously mentioned studies in the introduction.
Methods:
- Experimental design: why did the authors only include stage II and III diagnosed CRC cases?
- Plenty of description is missing for the sample processing and glycan analysis: which CRC tissue fragments were processed for protein extraction? What was the method for sonication? What was used for reduction and alkylation of proteins? What was the condition of the PGNAse F treatment (concentration of enzyme, time and temperature of incubation etc)? How was sequential elution in 5% acetic acid performed? How was trypsin digestion performed?
- The authors normalized the glycan ratios by the area ratio of the bi-antennary glycan HexNAc4Hex5NeuAc2. The authors need to show evidence that this glycan is not changing between CRC and Healthy conditions otherwise the relative comparison across samples will be biased. Alternative method for normalization by the total sum could be performed instead.
Results:
- The authors need to provide MS/MS support for ALL the glycan structures drawn in this study. Some structures are very uncommon in serum/plasma such as multi-fucosylated structures. Importantly, NeuGc glycans should not be expected in human serum and tissue and need to be supported by strong evidence if reported. Plenty of literature is available to support or not the structures found in this study. I would particularly recommend that the author check glycomics studies that used PGC-LC-MS/MS for glycan isomeric separation and characterization (e.g g Sethi et al, 2015, Glycobiology, https://doi.org/10.1093/glycob/cwv042, Chatterjee et al., 2021, J Clin Med, DOI: 10.3390/jcm10030516). The authors need to revise their glycomics data accordingly and relate their findings to the literature.
- The low overlap between MALDI-TOF-MS and LC-MS/MS results is a concern. The authors need to explain why such discrepancy was found.
- I suggest that the authors explore the results by grouping the glycans by major classes/features and compare the summed relative abundance across conditions.
- I am interested to know if there is any glycan change between different stage, cancer location, presence or not of mutation.
- I suggest that the author organize in a table or figure format a summary result comparison between the findings of the current study and previous studies performed in serum/plasma and tissue of CRC. It is important that the reader know what result is going in the same direction of the literature and what is not. Please include the references I mentioned above regarding the previous glycome studies in CRC tissue.
Minor:
- Line 71: add “variations in compositions and structures”
- Line 78: “Little is known about glycosylation”. This is not true. Plenty of literature is available showing the glycosylation changes in CRC and the potential of glycosignatures for CRC detection and prognosis.
- Line 85: ref 16 is completely misplaced. Please change it.
- Line 86: “total serum N-glycome (TSNG) has recently been described” but refences are from 2008 and 2016. Please include more recent references.
- Line 101: I could not understand what the authors mean by “consecutively operated colon cancer patients”
- Line 165: I find unusual to have an extraction buffer of “50% acetonitrile (ACN) solution in water” since acetonitrile in high concentration precipitate the proteins. Did the author include any other denaturing reagent? Were the samples dried after collection of the upper liquid phase and prior protein digestion? These are very critical points in the protocol that needs to be clearly described.
- Why did the authors use heavy iodomethane to label the pool reference in the serum N-glycome analysis but the same approach was not used for the tissue N-glycome analysis?
- Table 3: please state if this is from serum or tissue?
- Figure 5: please add a base line at the log ratio 0 to better visualize what is above or below 0.
- Line 437: glycan 17 should be HexNAc2Hex7 not HexNAc5Hex2. Glycan 15 should be HexNAc5Hex3, not HexNAc5Hex2.
- Line 502: the word “exclusively” is not accurate.
- The word “truncated” is not correctly used in this study. What is the definition of truncated used by the authors? In the literature truncated glycans are often referred as short glycans composed by the core N-glycans and truncated forms, e.g HexNAc1-2Fuc0-1, HexNAc2Hex1-3Fuc0-1 (paucimannose glycans).
Author Response
Manuscript ID: Biology-1123802
Identification of Differential N-Glycan Compositions in the Serum and Tissue of Colon Cancer Patients by Mass Spectrometry
Biology, section Medical Biology
Prof. Dr. Chris O'Callaghan (Editor-in-Chief)
Reviewer board
Biology
Dear Prof. Dr. Chris O'Callaghan and reviewer board,
Thank you for considering our manuscript titled “Identification of Differential N-Glycan Compositions in the Serum and Tissue of Colon Cancer Patients by MassSpectrometry” for publication on Biology.
We have had the opportunity to review our manuscript and have made the relevant changes as suggested. We re-affirm that the authors listed have fulfilled the requirements as set by the consensus statement on journal authorship. We report no conflicts of interests and have adhered to ethical standards for research.
We also reconfirm the authors list and affiliations, as presented before.
Please find below our point-by-point reply to major and minor modifications suggested by reviewers 1, 2 and 3.
Moreover, all modifications listed below are depicted on main manuscript text in bold type.
Accordingly, figures 3, 4 and 5; tables 2, 3 and 4 as well as supplementary material 1 (Table S2) and supplementary material 3 were fully rebuilt, as consequence of the major changes required.
In attention to the suggestion made by reviewer 1, we have created two tables in which we present our findings in comparison to the actual literature on N-glycosylation in CRC, provided as supplementary material 5.
Sincerely,
Marcelo de Melo Andrade Coura, MD, MSc
Dear reviewer 1,
Thank you very much for the thorough reading of the manuscript. Your considerations were important in the improvement of the quality of the manuscript. Please find below our point-by-point response, as requested.
Major:
Introduction:
- It is not clear what is the aim of the study. By starting the introduction describing the problem of low survival and its association with the cancer staging, the authors give the impression that a biomarker for CRC prognosis will be addressed in the study, which was not the case. Please provide a background description of the current gold standard diagnostic and prognostic methods in CRC used in clinics and what is the gap that the authors aim to fill.
The following sentences were added to the introduction:
Therefore, early detection of CRC is key. Currently, colonoscopy is the gold standard for diagnostic purposes in CRC, and albeit highly accurate, this is an invasive method, not exempt from complications. In turn, many biomarkers currently in use, mostly carbohydrate antigens such as CEA and CA 19-9, show low sensitivity and specificity in CRC early detection.
After surgical resection, surveillance is recommended for CRC patients to early detection of recurrent disease. Even though many patients will receive adjuvant chemotherapy some of them will not benefit from treatment, eventually developing metastasis.
In addition, the aim of the study was made clearer, by the addition of the following sentence:
Nevertheless, previous MS-based studies for biomarker discovery have addressed the N-glycome in serum or the N-glycome in tissues of CRC patients separatedly, as one assumes that compositions found in blood stream may reflect glycosylation present in the tumour microenvironment. In this study we analyzed, both serum and tissue N-glycome of CRC patients, by two MS-based technologies MALDI/MS and LC-MS/MS what revealed a special serum pattern compared to healthy individuals. More importantly, N-glycome in serum of CRC patients was highly different from compositions observed in tumour tissues.
- The authors missed some relevant studies that profile the changes in the N-glycome of CRC tissues (e.g Sethi et al, 2015, Glycobiology, https://doi.org/10.1093/glycob/cwv042, Kaprio et al., Mol Cell Proteomics, DOI:https://doi.org/10.1074/mcp.M114.041632, Boyaval, et al., 2021, Mol Cell Proteomics, DOI: 10.1074/mcp.RA120.002215).
The following sentence was added to the introduction:
Recently, mass spectrometry imaging (MSI)-based study in stage II CRC has revealed spatial distribution of N-glycans in tumour microenvironment with increase of mannose rich and sialylation levels in stroma, spreading into tumour invasive front [21].
- The authors need to state what is the main difference between the current study and the previously mentioned studies in the introduction.
The following sentence was added in the end of the introduction:
Nevertheless, previous MS-based studies for biomarker discovery have addressed the N-glycome in serum or the N-glycome in tissues of CRC patients separately, as one assumes that compositions found in blood stream may reflect glycosylation present in the tumour microenvironment. In this study we analyzed, both serum and tissue N-glycome of CRC patients, by two MS-based technologies MALDI/MS and LC-MS/MS what revealed a special serum pattern compared to healthy individuals. More importantly, N-glycome in serum of CRC patients was highly different from compositions observed in tumour tissues.
Methods:
- Experimental design: why did the authors only include stage II and III diagnosed CRC cases?
We did not include stage I and stage IV cases as stage I, no extramural disease, might not elicit a proper inflammatory response to tumour tissue, in that case with no power to significantly modify N-glycome profile in serum of cancer patients related to healthy controls. Conversely, by including stage IV cases, metastatic disease, the subversion of N-glycome in serum of cancer patients would surely be much more intense, however, with few clinical applications, as for diagnostic or prognostic purposes.
In fact, our group is devising a new protocol in which these groups will be included as well as a separate set of rectal cancer cases, in a larger cohort of CRC patients and healthy controls.
- Plenty of description is missing for the sample processing and glycan analysis: which CRC tissue fragments were processed for protein extraction? What was the method for sonication? What was used for reduction and alkylation of proteins? What was the condition of the PGNAse F treatment (concentration of enzyme, time and temperature of incubation etc)? How was sequential elution in 5% acetic acid performed? How was trypsin digestion performed?
The following sentences were added to the methods section:
Please find in 2.3.2. Tissue
Only distal fragments were processed for tissue extraction. Proximal samples were stored as repository for further analysis when deemed necessary
Please find in 2.4. protein extraction from tissue samples
For protein extractions, 60 mg of powdered tissue were dissolved in 500 µL of a 50% acetonitrile (ACN) solution in water, vortexed for 1 min, extracted in a bath sonicator for 45 min at room temperature and subsequently centrifuged at 14,000 rpm for 45 minutes. The upper liquid phase was collected and freeze dried for posterior glycoprotein extraction.
Please find in 2.5. N-glycan isolation and derivatization
For reduction, it was used 9.6 μl of dithiothreitol (DTT) 500mM and for alkylation 9.2 μl iodoacetoamide (IAA) 3M, in 200 μl of a buffer solution (pH 8.4).
PGNAse F treatment was carried out by adding to the digested glycopeptide samples, 3 μl of PNGAse F solution (1μg/μl) at 37°C, for 17h
Sep-Pak C18 columns were sequentially conditioned with 5 ml of methanol and 10 ml of 5% acid acetic solution. Samples containing free N-glycans and peptides were eluted in 200 μl of 5% acetic acid and added to the previously conditioned Sep-Pak columns. Sequential elutions were carried out by adding 3 ml of 5% acetic acid and 3 ml of 5% acetic acid with 80% acetonitrile. Free N-glycans were collected after elution with 3ml of 5% acetic acid.
Trypsin digestion was carried out by adding 14 μl of a trypsin solution (5 μg/μl) to reduced and alkylated glycoproteins for 24h, at 37°C
- The authors normalized the glycan ratios by the area ratio of the bi-antennary glycan HexNAc4Hex5NeuAc2. The authors need to show evidence that this glycan is not changing between CRC and Healthy conditions otherwise the relative comparison across samples will be biased. Alternative method for normalization by the total sum could be performed instead.
Results:
- The authors need to provide MS/MS support for ALL the glycan structures drawn in this study. Some structures are very uncommon in serum/plasma such as multi-fucosylated structures. Importantly, NeuGc glycans should not be expected in human serum and tissue and need to be supported by strong evidence if reported. Plenty of literature is available to support or not the structures found in this study. I would particularly recommend that the author check glycomics studies that used PGC-LC-MS/MS for glycan isomeric separation and characterization (e.g g Sethi et al, 2015, Glycobiology, https://doi.org/10.1093/glycob/cwv042, Chatterjee et al., 2021, J Clin Med, DOI: 10.3390/jcm10030516). The authors need to revise their glycomics data accordingly and relate their findings to the literature.
It is technically not possible to provide MS/MS spectra for all ions included in the present study. Several ions were not sufficiently intense to be fragmented and produce reliable MS/MS spectra, and therefore, for these ions, glycan compositions were assigned based solely in accurate MS analysis. Indeed, N-glycan assignments were performed using the GRITS toolbox, and the mass error for the precursor ion was set to < 5 ppm. Nevertheless, we concur with the reviewer that the assignment of N-glycan composition to ions based solely in MS measurements is fragile. Therefore, although all ions matching N-glycan compositions were monitored, we excluded from quantitative measurements ions for which we could not acquire MS/MS spectra. The following sentence was added to the text to express these changes:
Given that multiple ions did not produce reliable MS/MS spectra due to low signal intensity, only ions for which tandem MS spectra were obtained either for the permethylated or perdeuterated forms were considered in the development of quantitation methods.
In addition, changes were made in Table 3, and Figures 3, 4 and 5, in which we excluded all ions whose identification was performed based on accurate MS measurements alone. Also, additional changes reflecting the amount of evaluated glycans were made in the body of the manuscript.
- The low overlap between MALDI-TOF-MS and LC-MS/MS results is a concern. The authors need to explain why such discrepancy was found.
Signal suppression is a very well-known phenomenon in mass spectrometry. MALDI measurements are even more prone to such effects in relation to LC-MS, given the chromatographic separation of glycans. The lack of correlation between MALDI and LC-MS measurements is indeed puzzling, and in our understanding can be attributed mostly to signal suppression. However, this can be elaborated further:
- For serum N-glycan analyses, most of the compositions which were found to differ significantly between control and CRC patients resulted from ions of low abundance, which are more likely to suffer from ion suppression effects. One of the most relevant findings in our study is that HexNAc4Hex5Fuc1NeuAc2 (Glycan 58) is augmented in CRC patients, as detected by LC-MS analyses. The same tendency was observed in MALDI analyses, however, it was not statistically significant:
It is also noteworthy that other studies that uncover alterations in N-glycan levels in the serum of CRC patients by MALDI MS often rely in more sophisticated statistical analyses, such as partial least squares-discriminant analysis (PLS-DA) to uncover variations between controls and patients (10.18632/oncotarget.25753). This analysis provides scores that maximize the separation between pre-defined classes (controls vs disease). It is therefore likely that alterations in serum levels of N-glycans are too subtle to be captured by MALDI MS, and can only be identified after the application of statistical models.
- For tissue analysis, signal suppression was, in our understanding, even more relevant. Heterogeneity in the MALDI MS spectra obtained for different patients was thoroughly reported in the manuscript, and the spectra have been attached as supplementary material. Therefore, in our understanding, obtaining reliable data by MALDI MS from these set of spectra was challenging and we restricted ourselves to a more conservative approach. This resulted in low correlation with LC-MS data.
To make clearer that we understand that signal suppression is the main motive behind the lack of correlation between MALDI MS and LC-MS/MS measurements, we added these lines to the main text of the manuscript, in discussion section:
Albeit the number of compositions found in our MALDI/MS analysis are in line with previous MALDI/MS-based studies [ 15,41,52], lack of agreement between MALDI/MS and LC-MS/MS measurements is noteworthy. In contrast to LC-MS/MS analysis, MALDI/MS analysis yielded only three compositions capable of discriminating CRC patients from healthy controls, in both serum and tissue samples. In our understanding, ion signal suppression was responsible for this discrepancy. In this sense, we observed that some compositions may have not reached statistical significance merely due to its low ion intensity in samples, the ones more likely to suffer the influence from signal suppression.
- I suggest that the authors explore the results by grouping the glycans by major classes/features and compare the summed relative abundance across conditions.
We would like to thank the reviewer for the suggestion. Nevertheless, we developed a stable isotope labeling method to express the relative quantities of glycans based on the perdeuterated nearest neighbor. This analysis is an alternative to the summed relative abundance of ions and does not produce a straightforward way to normalize multiple glycans based in major classes/features.
- I am interested to know if there is any glycan change between different stage, cancer location, presence or not of mutation.
The following sentence was added to the body of the manuscript, in discussion section, as suggested:
Albeit highly heterogenous, not a single composition was found consistently over- or under-expressed when tumour tissue samples were stratified by primary CRC location, i. e., right- or left-sided, stage or mutational status (data not shown). These findings are in line with Holm et al study, in which no difference was found in non-acidic N-glycans by stage or primary location, even though a N-glycan signature in tumour stroma has been recently attributed to stage II CRC samples [18,20].
- I suggest that the author organize in a table or figure format a summary result comparison between the findings of the current study and previous studies performed in serum/plasma and tissue of CRC. It is important that the reader know what result is going in the same direction of the literature and what is not. Please include the references I mentioned above regarding the previous glycome studies in CRC tissue.
Please note that both tables will be submitted as supplementary material 5, cited in main text in the discussion section
SERUM
Table S3. N-glycosylation modifications in serum of CRC patients
|
Method |
Aiming |
Finding |
Composition |
Reference |
|
Lectin blot |
Total serum N-glycans, immunodepleted plasma,CRC patients vs controls |
Increase of sialylation and fucosylation |
↑ |
Qiu et al, 2008 |
|
Aleuria aurantia lectin blot |
β-haptoglobin N-glycosylation, CRC patients, Chron's disease and controls |
Higher AAL affinity in haptoglobin from CRC patients |
↑ |
Park et al, 2010 |
|
Lectin blot |
Total serum N-glycans, CRC patients vs controls |
Decrease of core fucosylation |
↓ |
Zhao et al, 2011 |
|
Liquid chromatography (UPLC) |
IgG N-glycosylation, CRC patients vs controls |
Decrease of galactosylation and sialylation, increase of core fucosylation |
↓ ↑ |
Vučković et al, 2016 |
|
Liquid chromatography (UPLC) |
IgG N-glycosylation, Prognostic in CRC patients |
Higher mortality with decrease of galactosylation and sialylation Increase of bisecting forms |
↓ ↑ |
Theodoratou et al, 2016 |
|
MALDI-TOF/MS, electrophoresis |
Total serum N-glycans, CRC patients vs controls |
Increase of multi-antennae core- and outer-arm fucosylated |
↑ |
Snyder et al, 2016 |
|
Liquid chromatography (UPLC) |
Total serum N-glycans, CRC patients vs controls |
Decreased of core-fucosylated di-antennary asialo and monosialo, Increase of multi-antennae sialylated |
↓ ↑ |
Doherty et al, 2018 |
|
MALDI-TOF/MS |
Total serum N-glycans, prognostic in CRC patients |
Decrease of core-fucosylated di-antennary Increase of multi-antennae sialylated, sialyl Lewis |
↓ ↑ ↑ |
de Vroome et al, 2018 |
|
MALDI-TOF/MS, LC/MS |
Total serum N-glycans, CRC patiens vs controls |
Increase of mannose -rich, bianntenary core fucosylated di-sialo and multi-antennae sialylated forms decrease of galactosylated forms |
↑ ↑ ↑ ↓ |
present study, 2021 |
TISSUE
Table S4. N-glycosylation modifications in cell lines and tissues of CRC
|
Method |
Aiming |
Finding |
Compositions |
Reference |
||||
|
CRC tissues/Lectin |
N-glycosylation profile in MUC1/CEACAM 5 glycoproteins |
Increase of sialylated, high-mannose and branched N-glycans in CEACAM 5 |
Saeland et al, 2012 |
|||||
|
CRC tissues/ MALDI-TOF/MS, HILIC CHROMATOGRAPHY |
Comparison between CRC tumour and paired normal tissues |
Increase of high-mannose, sulfated, paucimannosidic and sLewis X(sLex), decrease of bisecting compositions in tumour |
|
Balog et al, 2012 |
||||
|
CRC tissues/Lectin blot |
Comparison between CRC tumour and paired normal tissues |
Increase of α2,3 sialylated residues |
|
Fukasawa et al, 2013 |
||||
|
CRC tissues and cell lines/LC-MS |
Comparison between CRC tumour and cell lines (SW1116, SW480, SW620, SW837, LS174) |
Increase of high-mannose in tumour and cell lines |
Chik et al, 2014 |
|||||
|
Cell lines/MS |
N-glycans analysis of three cell lines (LIM 1215, LIM 1819 and LIM2405) |
Increase of high-mannose and α2,6-sialylated N-glycans in all cell lines. Increase of bisecting compositions in LIM 1215 and increase of α2,3 sialylated compositions in LIM2405 |
|
Sethi et al, 2014 |
||||
|
CRC tissues/MALDI-TOF/MS |
Comparison of N-glycans among tumour tissue, adenoma and paired normal tissue of rectal tumours |
Increase of small mannose, paucimannosidic and sialylated compositions in CRC tissue in relation to adenoma. Worst prognosis in tumours with increased paucimannosidic |
Kaprio et al, 2015 |
|||||
|
CRC tissues/
|
Comparison between N-glycans in EGFR + and EGFR -tumour tissues with paired normal tissues
|
Increase of high-mannose, paucimannosidic, hybrid compositions and higher α2,6- sialylation.
High bisecting and low α2,3 sialylation in EGFR + |
|
Sethi et al, 2015
|
||||
|
CRC tissues/Lectin array |
N-glycan profile in CEA of tumour tissues compared no paired normal tissues |
Increase of fucose and mannose residues, decrease of branched and bisecting compositions in tumours. Decrease of mannose, galactose, N-Acetylglucosamine and N-Acetylgalactosamine in more advanced disease |
|
Zhao et al, 2018 |
||||
|
CRC tissues/LC-MS |
Comparison between N-glycosylation profile in tumour and paired normal tissues |
Increase of high-mannose and bi-fucosylated and decrease of bisecting compositions in tumour. Decrease of bisecting compositions in more advanced disease |
|
Zhang et al, 2019 |
||||
|
CRC tissues/MALDI-TOF/MS |
N-glycans profile in right- and left- sided tumour tissues and normal colon tissues of healthy individuals |
Increase of acidic, paucimannosidic, high mannose N-glycans and decrease of bisecting compositions in tumour samples. No difference related to stage or sidedness |
|
Holm et al, 2020 |
||||
|
CRC tissue/MALDI imaging |
Samples of stage II CRC tissue and peritumoral tissue |
Increase of high mannose N-glycans and sialylation
Decrease of fucosylation and branched N-glycans
|
|
Boyaval et al, 2020 |
||||
|
CRC tissue/MALDI-TOF/MS LC/MS |
Comparison between N-glycosylation profile in tumour and paired normal tissues |
Increase of high mannose, paucimannosidic, bi-antennary mono-galctosylated and branched N-glycans |
Present study, 2021 |
Minor:
Line 71: add “variations in compositions and structures”
Please find edited in the main text, as suggested.
- Line 78: “Little is known about glycosylation”. This is not true. Plenty of literature is available showing the glycosylation changes in CRC and the potential of glycosignatures for CRC detection and prognosis.
Please find edited in the main text, as suggested.
- Line 85: ref 16 is completely misplaced. Please change it.
Please find edited in the main text (changed to ref 22)
- Line 86: “total serum N-glycome (TSNG) has recently been described” but refences are from 2008 and 2016. Please include more recent references.
This sentence was modified to:
More importantly, the human total serum N-glycome (TSNG) has comprehensively been described and, albeit variations amongst normal individuals, it is believed that many pathologic states including inflammation and cancer promote significant modifications on glycoproteins, easily identifiable from healthy individuals
- Line 101: I could not understand what the authors mean by “consecutively operated colon cancer patients”
This means that patients with colon cancer were operated on in sequence from October 2017 to June 2018. Nonetheless, the term "consecutively" was withdrawn from the text, in sake of clarity, without compromising the results of the study.
- Line 165: I find unusual to have an extraction buffer of “50% acetonitrile (ACN) solution in water” since acetonitrile in high concentration precipitate the proteins. Did the author include any other denaturing reagent? Were the samples dried after collection of the upper liquid phase and prior protein digestion? These are very critical points in the protocol that needs to be clearly described.
Before implementing the method for glycoprotein extraction, we carried out a pilot study for glycoprotein extraction that used 1 % SDS and 0,5% 2-mercaptoethanol solutions (doi: 10.1111/j.1742-4658.2005.05068.x)
During our initial analysis we experienced enormous difficulty in obtain sound acquisitions in MALDI/MS, especially for tissue samples, not mentioning have to add another step to remove detergents from samples.
Curiously, as we changed to 50% ACN solution followed by reduction with dithiothreitol (DTT) and alkylation with iodoacetoamide (IAA) in buffer solution (pH 8.4), we significantly improved our spectra quality, without compromising ion signals yielded.
Moreover, our protocol is based on a previous protocol for MS analysis of N- and O-glycans extracted from biological samples (Nature protocols, doi: 10.1038/nprot.2007.227 ). In this protocol the authors considered unnecessary to add SDS or other detergent analysis of samples containing > 50 μg of glycoproteins.
Furthermore, as we carried out our analysis on fresh frozen tissues, it was our assumption to reduce at a minimum the interference of extraction buffers, salt reagents and detergents on N-glycans freed from glycoproteins.
Yes, they were dried as stated in 2.4. protein extraction from tissue samples, Methods section.
- Why did the authors use heavy iodomethane to label the pool reference in the serum N-glycome analysis but the same approach was not used for the tissue N-glycome analysis?
It was not possible to obtain tissue fragments of control patients due to ethical reasons. Indeed, in contrast to serum, which is easily obtained from healthy donors, surgical procedures in healthy controls are unjustified. Therefore, we created an alternative approach, which was to compare the pool of proteins from adjacent normal tissue, as qualified by histological analysis, to CRC tissue samples, creating an index of altered proteins in CRC tissues for each patient. The analysis was then based on the consistency of the findings.
- Table 3: please state if this is from serum or tissue?
Please find edited in the main text
- Figure 5: please add a base line at the log ratio 0 to better visualize what is above or below 0.
Figure 5 was modified as suggested by the reviewer.
- Line 437: glycan 17 should be HexNAc2Hex7 not HexNAc5Hex2. Glycan 15 should be HexNAc5Hex3, not HexNAc5Hex2.
Please find edited in the main text
- Line 502: the word “exclusively” is not accurate.
Please find edited in the main text
- The word “truncated” is not correctly used in this study. What is the definition of truncated used by the authors? In the literature truncated glycans are often referred as short glycans composed by the core N-glycans and truncated forms, e.g HexNAc1-2Fuc0-1, HexNAc2Hex1-3Fuc0-1 (paucimannose glycans).
In sake of clarity the term "truncated" was changed for hypogalactosylated in simple summary and in abstract sections. Mono-galactosylated and mono-galactosylated caped with one sialic acid were used in the end of the discussion section.
Please find them bold typed in the main text as stated above

Reviewer 2 Report
In the manuscript “Identification of Differential N-Glycan Compositions in the Serum and Tissue of Colon Cancer Patients by Mass Spectrometry” Coura et al. use a combination of two mass spectrometry methods to analyse the N-glycome of colorectal cancer (CRC) patient derived samples. The manuscript is technically sound and provides a comparison on the abundance of serum N-glycans of healthy donors and 13 colorectal cancer patients. In addition, the authors evaluated also primary tumour colorectal tumours using comparable approaches. Overall, the authors provide a comprehensive repository on identified glycan alterations in sera and tumours of CRCs that may prove useful for future studies. Complex biological samples derived from patients are by nature very heterogenous hampering the conclusions that can be drown by purely descriptive studies. However, the authors discuss this issue sufficiently in this manuscript and place their findings in context to other more functional studies.
Minor critique:
- The authors should try to incorporate more recent publications in their introduction and discussion.
- Figure 3: Glycan 85 and 86. Are the authors absolutely certain about the bisecting GlcNAc and it being galactosylated? Please refrain from representing the detailed structure if not certain. The use of brackets like for glycan 62 for instance would be more appropriate in that case.
- Lines 67 to 69: “Approximately 60% of all human proteins are glycosylated, with N-glycosylation being the most specific and predictable type, differently than O-glycosylation”. Please rephrase this over-generalized statement.
- Lines 78 to 80: “Even though aberrant glycosylation is a hallmark of cancer development and progression [10], little is known about glycosylation as a phenotypical marker in colon cancer. Over the last 10 years, many studies have focused on the role of N-glycosylation in CRC.” The two statements appear to be contradicting.
- The meaning of the last paragraph (conclusions) is unclear and very speculative, and therefore not a justified conclusion of the manuscript: “Moreover, a subset of compositions upregulated in tissues may serve as cancer biomarker what needs confirmation in a large cohort of CRC patients, perhaps tested sole or in association with another serum marker, as CEA, thereby overcoming limitations in specificity for diagnostic purposes.”
Author Response
Manuscript ID: Biology-1123802
Identification of Differential N-Glycan Compositions in the Serum and Tissue of Colon Cancer Patients by Mass Spectrometry
Biology, section Medical Biology
Prof. Dr. Chris O'Callaghan (Editor-in-Chief)
Reviewer board
Biology
Dear Prof. Dr. Chris O'Callaghan and reviewer board,
Thank you for considering our manuscript titled “Identification of Differential N-Glycan Compositions in the Serum and Tissue of Colon Cancer Patients by MassSpectrometry” for publication on Biology.
We have had the opportunity to review our manuscript and have made the relevant changes as suggested. We re-affirm that the authors listed have fulfilled the requirements as set by the consensus statement on journal authorship. We report no conflicts of interests and have adhered to ethical standards for research.
We also reconfirm the authors list and affiliations, as presented before.
Please find below our point-by-point reply to major and minor modifications suggested by reviewers 1, 2 and 3.
Moreover, all modifications listed below are depicted on main manuscript text in bold type.
Accordingly, figures 3, 4 and 5; tables 2, 3 and 4 as well as supplementary material 1 (Table S2) and supplementary material 3 were fully rebuilt, as consequence of the major changes required.
In attention to the suggestion made by reviewer 1, we have created two tables in which we present our findings in comparison to the actual literature on N-glycosylation in CRC, provided as supplementary material 5.
Sincerely,
Marcelo de Melo Andrade Coura, MD, MSc
Dear reviewer 2,
Thank you very much for the thorough reading of the manuscript. Your considerations were important in the improvement of the quality of the manuscript. Please find below our point-by-point response, as requested.
Minor critique:
- The authors should try to incorporate more recent publications in their introduction and discussion.
Please find in introduction section:
Recently, mass spectrometry imaging (MSI)-based study in stage II CRC has revealed spatial distribution of N-glycans in tumour microenvironment with increase of mannose rich and sialylation levels in stroma, spreading into tumour invasive front [21].
- Boyaval, et al., 2021, Mol Cell Proteomics, DOI: 10.1074/mcp.RA120.002215
Please find in introduction section:
More importantly, the human total serum N-glycome (TSNG) has comprehensively been described and, albeit variations amongst normal individuals, it is believed that many pathologic states including inflammation and cancer promote significant modifications on glycoproteins, easily identifiable from healthy individuals
Please find in discussion section:
Albeit highly heterogenous, not a single composition was found consistently over- or under-expressed when tumour tissue samples were stratified by primary CRC location, i. e., right- or left-sided, stage or mutational status (data not shown). These findings are in line with Holm et al study, in which no difference was found in non-acidic N-glycans by stage or primary location, even though a N-glycan signature in tumour stroma has been recently attributed to stage II CRC samples [19, 21].
- Figure 3: Glycan 85 and 86. Are the authors absolutely certain about the bisecting GlcNAc and it being galactosylated? Please refrain from representing the detailed structure if not certain. The use of brackets like for glycan 62 for instance would be more appropriate in that case.
The reviewer is correct. The authors have checked all proposed structures within the manuscript for inconsistencies. Indeed, following the suggestion of another reviewer, all structures for which MS/MS spectra could not be obtained due to low signal intensity were removed from further analyses. Modifications were made in Figure 3, 4 and 5 as well as the body of the manuscript to accommodate such changes.
- Lines 67 to 69: “Approximately 60% of all human proteins are glycosylated, with N-glycosylation being the most specific and predictable type, differently than O-glycosylation”. Please rephrase this over-generalized statement.
Please find in introduction section:
Approximately 60% of all human proteins are glycosylated, with N-glycosylation occurring when a glycan is attached to asparagine (Asn) in the motif Asn- X- Ser/Thr, where X is not a proline residue
- Lines 78 to 80: “Even though aberrant glycosylation is a hallmark of cancer development and progression [10], little is known about glycosylation as a phenotypical marker in colon cancer. Over the last 10 years, many studies have focused on the role of N-glycosylation in CRC.” The two statements appear to be contradicting.
For the sake of consistency, the first sentence was modified to
Aberrant glycosylation has been acknowledged as a hallmark of cancer, resulting from alterations in the machinery of tumour cells, albeit it is not clear if this phenotypical modification is cause or consequence of malignant transformation
The second sentence was maintained
Over the last 10 years, many studies have focused on the role of N-glycosylation in CRC.
- The meaning of the last paragraph (conclusions) is unclear and very speculative, and therefore not a justified conclusion of the manuscript: “Moreover, a subset of compositions upregulated in tissues may serve as cancer biomarker what needs confirmation in a large cohort of CRC patients, perhaps tested sole or in association with another serum marker, as CEA, thereby overcoming limitations in specificity for diagnostic purposes.”
In conclusion, our study revealed a global decrease of galactosylated compositions present in serum of CRC patients by LC/MS technology. Conversely, three compositions were upregulated in serum pointing to a subset of structures highly discriminative between cases from healthy controls. Moreover, our results suggest this panel holds potential in clinical setting with applications in large scale by automated MS analysis.
Albeit heterogeneous, a commonality of high mannose and branched compositions could be identified upregulated in tumour tissues. However, some new compositions found increased in tumour, as oligomannosidic and bi-antennary mono-galactosylated glycans, have hardly been described in CRC, what will surely deserve closer attention in further studies. Curiously, no single N-glycan increased in tissues was found upregulated in serum, what poses the question where serum compositions in CRC patients are from.

Reviewer 3 Report
This is an interesting research underlining the role of N-Glycan compositions in the serum and tissue of colon cancer patients. Some minor issues are easy to fix to improve the quality of the comprehensive review.
Due to the number of samples is not large, I recommend to measures of accuracy that use the ROC curve in statistical analysis.
Please modify the conclusions. Please add few more key findings here.
I believe the paper is worth to be published.
Author Response
Manuscript ID: Biology-1123802
Identification of Differential N-Glycan Compositions in the Serum and Tissue of Colon Cancer Patients by Mass Spectrometry
Biology, section Medical Biology
Prof. Dr. Chris O'Callaghan (Editor-in-Chief)
Reviewer board
Biology
Dear Prof. Dr. Chris O'Callaghan and reviewer board,
Thank you for considering our manuscript titled “Identification of Differential N-Glycan Compositions in the Serum and Tissue of Colon Cancer Patients by MassSpectrometry” for publication on Biology.
We have had the opportunity to review our manuscript and have made the relevant changes as suggested. We re-affirm that the authors listed have fulfilled the requirements as set by the consensus statement on journal authorship. We report no conflicts of interests and have adhered to ethical standards for research.
We also reconfirm the authors list and affiliations, as presented before.
Please find below our point-by-point reply to major and minor modifications suggested by reviewers 1, 2 and 3.
Moreover, all modifications listed below are depicted on main manuscript text in bold type.
Accordingly, figures 3, 4 and 5; tables 2, 3 and 4 as well as supplementary material 1 (Table S2) and supplementary material 3 were fully rebuilt, as consequence of the major changes required.
In attention to the suggestion made by reviewer 1, we have created two tables in which we present our findings in comparison to the actual literature on N-glycosylation in CRC, provided as supplementary material 5.
Sincerely,
Marcelo de Melo Andrade Coura, MD, MSc
Dear reviewer 3,
Thank you very much for the thorough reading of the manuscript. Your considerations were important in the improvement of the quality of the manuscript. Please find below our point-by-point response, as requested.
- This is an interesting research underlining the role of N-Glycan compositions in the serum and tissue of colon cancer patients. Some minor issues are easy to fix to improve the quality of the comprehensive review.
Due to the number of samples is not large, I recommend to measures of accuracy that use the ROC curve in statistical analysis.
To create a ROC curve, one has to create a binary classifier for the data. In our report, we performed only the K-means clustering of the data and reported it as a constellation plot. Please see that this is an unsupervised approach to the data and therefore this does not constitute a binary classifier, as information about classes (controls vs CRC) is given. We refrained from presenting a binary classifier in the present report due to the sample size. This will be presented in the future when we expand the number of analyzed subjects and validate the findings presenter herein.
Nevertheless, following the reviewer´s suggestion, we performed a linear discriminant analysis using the stepwise method and created a ROC curve, as shown in the next page. It is clear that if one includes 3 N-glycans, HexNAc2Hex7 (Nr 17), HexNAc5Hex4 (Nr 22) and HexNAc4Hex5Fuc1NeuAc2 (Nr 58), the 3 compositions to present highest F-ratios, in the discriminant analysis matrix, 100% efficiency in the classification of control vs CRC patients is obtained. The ROC curve is demonstrated below.
- Please modify the conclusions. Please add few more key findings here.
Please find in conclusion section the following text
In conclusion, our study revealed a global decrease of galactosylated compositions present in serum of CRC patients by LC/MS technology. Conversely, three compositions were upregulated in serum pointing to a subset of structures highly discriminative between cases from healthy controls. Moreover, our results suggest this panel holds potential in clinical setting with applications in large scale by automated MS analysis.
Albeit heterogeneous, a commonality of high mannose and branched compositions could be identified upregulated in tumour tissues. However, some new compositions found increased in tumour, as oligomannosidic and bi-antennary mono-galactosylated glycans, have hardly been described in CRC, what will surely deserve closer attention in further studies. Curiously, no single N-glycan increased in tissues was found upregulated in serum, what poses the question where serum compositions in CRC patients are from.

Round 2
Reviewer 1 Report
This study has major flaws in the aims and experimental design. Although the authors now stated that biomarkers for early detection is key, early stage I was not included in the cohort of patients. The authors claimed in the response letter that “we did not include stage I and stage IV cases as stage I, no extramural disease, might not elicit a proper inflammatory response to tumour tissue, in that case with no power to significantly modify N-glycome profile in serum of cancer patients related to healthy controls”. However, no reference was provided for this statement. Besides, previous reports have documented changes in the glycome of plasma and tissue from CRC patients at early stages (e.g Kaprio et al., Mol Cell Proteomics, 2015 and Doherty et al, Scientific Reports, 2018). The low overlap between MALDI-MS and LC-MS/MS is still a concern to me. The explanation provided by the authors that one of the reasons for this discrepancy was the signal suppression, especially in MALDI did not convince me why the authors did not find similar glycan changes between cancer vs non-cancer using both methods. Were the glycans identified by MALDI-MS different from the glycans identified by LC-MS? The authors also failed in giving an answer about the evidence that the glycan HexNAc4Hex5NeuAc2 used for normalization of the glycome data is not changing between CRC and healthy conditions. Multifucosylated glycans structures are still proposed in the study without spectra evidence. Finally, the comparison between serum vs tissue only based on glycome data does not provide any relevant information. The glycoproteome is complex and it is impossible to infer any potential cellular origin without the knowledge of the glycoprotein carrier.
Author Response
Manuscript ID: Biology-1123802
Identification of Differential N-Glycan Compositions in the Serum and Tissue of Colon Cancer Patients by Mass Spectrometry
Biology, section Medical Biology
Prof. Dr. Chris O'Callaghan (Editor-in-Chief)
Reviewer board
Biology
Dear Prof. Dr. Chris O'Callaghan and reviewer board,
Thank you for considering our manuscript titled “Identification of Differential N-Glycan Compositions in the Serum and Tissue of Colon Cancer Patients by Mass Spectrometry” for publication on Biology.
We have had the opportunity to review our manuscript, once again, and have made the relevant changes as suggested. We re-affirm that the authors listed have fulfilled the requirements as set by the consensus statement on journal authorship. We report no conflicts of interests and have adhered to ethical standards for research.
We also reconfirm the authors list and affiliations, as presented before.
Please find below our point-by-point reply to modifications suggested by reviewer #1.
Sincerely,
Marcelo de Melo Andrade Coura, MD, MSc
Dear reviewer #1,
Thank you very much for your time and work in reviewing our first round reply. Your considerations were important in the improvement of the quality of the manuscript. Please find below our point-by-point response, as requested.
Reviewer 1 round 2
Reviewer 1. This study has major flaws in the aims and experimental design. Although the authors now stated that biomarkers for early detection is key, early stage I was not included in the cohort of patients. The authors claimed in the response letter that “we did not include stage I and stage IV cases as stage I, no extramural disease, might not elicit a proper inflammatory response to tumour tissue, in that case with no power to significantly modify N-glycome profile in serum of cancer patients related to healthy controls”. However, no reference was provided for this statement. Besides, previous reports have documented changes in the glycome of plasma and tissue from CRC patients at early stages (e.gKaprio et al., Mol Cell Proteomics, 2015 and Doherty et al, Scientific Reports, 2018).
Our reply
Regarding to modifications in the N-glycome of tissues from CRC patients, Kaprio et al. identifies altered paucimannosidic glycans H3N2 and H4N2 in stage I CRC tissue. However, it is noteworthy that this study has focused only on rectal cases, while our study has included only colon cancers and excluded rectal cancer cases. Rectal cancer was excluded from our analysis, for two main reasons: firstly, due to the uncertainty of the modifications in N-glycosylation of two tissues with different embryological origin, as such proximal colon tissues (midgut) and rectal tissues (hindgut). Despite we included left-sided and right-sided colon cancer samples altogether in the analysis, we found not one composition differently expressed between samples regarding to primary tumor location, very similar to the results of HOLM (HOLM et al, PLoS One, 2020, doi:10.1371/journal.pone.0234989 ).
Secondly, since we have studied fresh samples of stage II and III tumors collected after the surgical resection, the addition of radio and chemotherapy pre-operatively, as part of their treatment (SAUER et al, NEJM, 2004, doi:10.1056/nejmoa040694), would have interfered on the N-glycome in rectal cancer tissues, as previously reported (LEE et al., IJROBP, 2010, doi.org/10.1016/j.ijrobp.2009.11.022).
Moreover, in the study of Kaprio et al, the authors have grouped stage I-II rectal cancer samples labeled as “local disease”, stage III as “locoregional disease” and stage IV as “advanced disease”. By doing that, they were able to clearly identify a characteristic profile of N-glycome in local disease, what might suggest that stage I and II stages have similar N-glycosylation profiles in rectal cancer, as opposed to the pre-cancerous adenoma stage, as well as in more advanced disease.
Doherty et al., using chromatography, reports that N-glycans (F(6)A2G(4)2) and (F(6)A2G(4)2 S(6)1) are markers of all stages of the disease in serum.
In this study, the authors described a decrease of N-glycans (F(6)A2G(4)2) and (F(6)A2G(4)2 S(6)1) associated with a progressive increase of the large group of consecutive peaks, named as “tail peak”, across all CRC stages. However, in stage I cases, they found no increase of peaks in this later retention times of the chromatogram. This region is likely comprised by tri-, tetra-antennary, galactosylated glycans lacking a core-fucose. They suggest this region is linked to markers of inflammation, as C-reactive protein (CRP), and since stage I tumours have not yet metastasized, out of the bowel wall, it might be possible that inflammation may not be present yet. Similar with our conclusions, the authors assumed that, as stage I disease evolves into more inflamed stages, so increases acute phase proteins in serum and with them increases tri-tetra sialylation/galactosylation.
Considering these two papers cited by the reviewer and numerous others in the doi: 10.1002/ijc.24685, doi: 10.1002/cncr.26342, doi: 10.1158/1078-0432.CCR-15-1867, doi: 10.1021/acs.analchem.6b02310, doi: 10.1038/srep28098, doi: 10.1038/s41598-018-26805-7, doi: 10.18632/oncotarget.25753, doi: 10.1074/mcp.M111.011601, doi: 10.1074/mcp.M114.041632, doi: 10.1093/glycob/cwv042, doi: 10.1093/glycob/cwz005/5304492, doi: 10.1186/s12014-018-9182-4, doi: 10.1371/journal.pone.0234989) it is possible to observe that there is still no unique marker for CRC and that there is a methodological bias in the identification of altered N-glycans in the serum of CRC patients. It is, therefore, necessary to develop putative markers for each methodology, such as the mass spectrometry-based approach presented in the present manuscript.The authors aimed at identifying putative markers in conditions where an inflammatory response is already detected, such as stage II CRC, to develop putative markers for future application in the early detection of stage I CRC. Indeed, as previously stated, stage I and stage IV CRC patients will be included in a future cohort, where the putative serum markers described herein will be further evaluated.
Reviewer 1. The low overlap between MALDI-MS and LC-MS/MS is still a concern to me. The explanation provided by the authors that one of the reasons for this discrepancy was the signal suppression, especially in MALDI did not convince me why the authors did not find similar glycan changes between cancer vs non-cancer using both methods. Were the glycans identified by MALDI-MS different from the glycans identified by LC-MS?
Our reply
Lack of correlation between MALDI and LC-MS/MS results was, primarily, a concern to us as well. To investigate the reasons for such, we randomly selected 2patients, extracted peak areas for some glycan structures, and calculated the Pearson correlation coefficient between areas of the most intense peaks and compared to the areas of less intense peaks from the same patients.
In SERUM:
Most intense peaks:
|
CRC 1 patient |
CRC 4patient |
||||
|
Glycan nr, Met or Deu |
LC-MS peakarea |
MALDI peakarea |
Glycan nr, Met or Deu |
LC-MS peakarea |
MALDI peakarea |
|
13_PerMet |
654816.6328 |
232 |
13_PerMet |
896331.1331 |
398 |
|
13-Deu |
548349.9543 |
181 |
13-Deu |
471963.7272 |
135 |
|
19_PerMet |
813345.5695 |
633 |
19_PerMet |
673704.8279 |
506 |
|
19-Deu |
519516.6937 |
224 |
19-Deu |
404794.8485 |
146 |
|
27-Met |
514735.555 |
419 |
27-Met |
215748.4464 |
116 |
|
35-Met |
946979.3534 |
931 |
35-Met |
731888.432 |
542 |
|
52-PerMet |
5843107.794 |
36247 |
44-Met |
317691.0401 |
13 |
|
52-Deu |
1110747.27 |
4574 |
51-Met |
284815.2708 |
183 |
|
58-PerMet |
370111.1265 |
1449 |
52-PerMet |
4414533.903 |
7226 |
|
58-Deu |
141332.7883 |
154 |
52-Deu |
1161676.817 |
1377 |
|
64-PerMet |
102811.3115 |
119 |
58-PerMet |
442593.1138 |
291 |
|
|
|
58-Deu |
109181.7655 |
51 |
|
|
|
|
|
|
||
|
|
R^2 |
0.978399194 |
|
R^2 |
0.982942975 |
Less intense peaks:
|
CRC 1 patient |
CRC 4patient |
||||
|
Glycan nr, Met or Deu |
LC-MS area |
MALDI area |
Glycan nr, Met or Deu |
LC-MS area |
MALDI area |
|
6-Met |
918184.8216 |
59 |
6-Deu |
1161676.817 |
15 |
|
6-Deu |
1111072.592 |
16 |
15-Met |
12136.9718 |
15 |
|
13-Met |
654816.6328 |
34 |
20-Met |
91470.66634 |
15 |
|
17-Met |
152740.4507 |
21 |
25-Met |
136007.7454 |
14 |
|
20-Met |
208110.9743 |
36 |
32-Met |
166830.2481 |
15 |
|
21-Met |
106879.2547 |
45 |
48-Met |
16268.29363 |
12 |
|
24-Met |
123063.7792 |
12 |
64-Deu |
109181.7655 |
18 |
|
25-Met |
227831.2532 |
21 |
65-Met |
72437.53754 |
22 |
|
28-Deu |
519516.6937 |
25 |
70-Met |
9562.458822 |
19 |
|
33-Met |
420026.9203 |
33 |
79-Met |
10490.09758 |
10 |
|
49-Met |
22122.8788 |
17 |
80-Met |
37917.90177 |
4 |
|
64-Deu |
141332.7883 |
21 |
81-Met |
35217.06967 |
4 |
|
R^2 |
0.088801173 |
R^2 |
0.005793088 |
||
The same analysis was carried on with mass spectra obtained from CRC tissue:
CRC tissue:
Most intense peaks:
|
CRC 3 tissue |
CRC 5 tissue |
||||
|
Glycan nr, Met or Deu |
LC-MS area |
MALDI area |
Glycan nr, Met or Deu |
LC-MS area |
MALDI area |
|
6-PerMet |
26676.5513 |
234 |
27-Deu |
24380.95847 |
365 |
|
10-PerMet |
5301.730314 |
270 |
33-Deu |
8153.85688 |
421 |
|
13_PerMet |
147.6197755 |
60 |
44-Deu |
26082.37164 |
1017 |
|
17_PerMet |
60.10917158 |
123 |
52-Deu |
63180.21436 |
3011 |
|
25_PerMet |
24300.60115 |
85 |
58-Deu |
4574.755105 |
767 |
|
27-Met |
4152.161496 |
127 |
72-Deu |
745.1436798 |
343 |
|
44-Met |
44242.01326 |
217 |
|
||
|
44-Deu |
6200.356649 |
65 |
R^2 |
0.82201844 |
|
|
52-PerMet |
117847.8908 |
485 |
|
||
|
52-Deu |
55685.06766 |
355 |
|
||
|
58-PerMet |
3228.684178 |
143 |
|
||
|
58-Deu |
3197.91698 |
50 |
|
||
|
|
|
|
|||
|
|
R^2 |
0.744656666 |
|
||
Less intense peaks:
|
CRC 3 tissue |
CRC 5 tissue |
||||
|
Glycan nr, Met or Deu |
LC-MS area |
MALDI area |
Glycan nr, Met or Deu |
LC-MS area |
MALDI area |
|
7-Met |
1116.312051 |
16 |
13-Deu |
1866.486006 |
11 |
|
6-Deu |
11168.47823 |
34 |
19_PerMet |
147.4176524 |
9 |
|
13-Deu |
3061.583794 |
17 |
19-Deu |
2060.550651 |
13 |
|
19_PerMet |
48.16800415 |
42 |
17_PerMet |
80.78496494 |
36 |
|
19-Deu |
1463.957765 |
11 |
17-Deu |
3116.648321 |
7 |
|
17-Deu |
3806.511216 |
22 |
28-PerMet |
4414.478112 |
8 |
|
25-Deu |
3625.730225 |
23 |
28-Deu |
1051.26003 |
11 |
|
27-Deu |
2343.648373 |
26 |
52-Deu |
4070.596135 |
38 |
|
28-PerMet |
5438.613465 |
23 |
58-Deu |
1262.354927 |
0 |
|
28-Deu |
401.5271938 |
7 |
64-PerMet |
1650.561546 |
0 |
|
64-PerMet |
6131.428095 |
4 |
64-Deu |
205.6628 |
5 |
|
64-Deu |
417.8295549 |
9 |
72-PerMet |
1195.429465 |
13 |
|
|
|
|
|
||
|
|
|
|
|
||
|
|
R^2 |
0.053739001 |
|
R2 -intensos |
0.01797515 |
From these analyses we apprehend that:
- Correlation between peak areas obtained from MALDI and LC-MS measurements in SERUM is high when one considers the most intense peaks.
- Correlation between peak areas obtained from MALDI and LC-MS measurements fall to approximately zero when low intensity peaks are considered.
- Correlation between peak areas obtained from MALDI and LC-MS measurements in CRC TISSUE is high, but lower than those obtained for SERUM analyses. This is, in our understanding, due to the overall lower intensity of N-glycan ions found in these spectra and the high heterogeneity between MALDI spectra obtained for TISSUE N-glycan extracts, which often presented several other ions that do not match any N-glycan composition. This was thoroughly documented in the main body of the manuscript, and the MALDI spectra from CRC tissue was attached.
- Correlation between peak areas obtained from MALDI and LC-MS measurements in CRC TISSUE falls to approximately zero when only less intense ions are considered.
Overall, the good agreement between peak areas obtained from MALDI and LC-MS measurements for intense ions and the lack of agreement between less intense ions is, in our understanding, a consequence of signal suppression, as already written in the main manuscript body and reinforced in the previous response to the reviewer.
Reviewer 1. The authors also failed in giving an answer about the evidence that the glycan HexNAc4Hex5NeuAc2 used for normalization of the glycome data is not changing between CRC and healthy conditions.
Our reply
As stated in the previous response to the reviewer: “By calculating the area ratio of all N-glycans in relation to HexNAc4Hex5NeuAc2, we provide a normalized measure of glycan abundance within the serum of each patient. In our understanding, variable absolute concentrations of HexNAc4Hex5NeuAc2 between patients does not change the validity of using ratios for the diagnosis of diseases.”, the authors did not state that the glycan HexNAc4Hex5NeuAc2 is not changing between CRC and healthy controls. Our result shows that the peak areas do indeed change between subjects, regardless of their origin, if they are from CRC or control subjects. However, the use of HexNAc4Hex5NeuAc2 for the normalization of the remaining glycans is a valid approach for the differentiation between healthy and CRC subjects, as exemplified, in the previous response, for the diagnosis of MCADD.
Reviewer 1. Multifucosylated glycans structures are still proposed in the study without spectra evidence.
Our reply
We are still unsure to what structures the reviewer is referring to, as he/she has not singled out any. Please be more specific. Although multi-fucose structures are indeed uncommon in the plasma of patients, they have been previously described in proteins such as AGP (10.1016/S0009-8981(02)00427-8 and 10.1002/jms.938). Notice that all ions that were considered for quantitation presented MS/MS spectra for either the permetylated or perdeuterated forms, or both, as attached in the supplementary material (Supp mat 2 and 3) along with the report generated either by the GRITS or Glycoworkbench software.
Find below examples of the annotated spectra for glycans 67 (HexNAc6Hex7Fuc2) and 87 (HexNAc4Hex3Fuc2), the firstin permethylated form and the second, perdeuterated. For glycan 67, we identified ions that locate the fucose residues in the outer arms of the structure and not in the chitobiose core. Also, it is important to stress that this glycan, although evaluated in serum, was considered altered in the cancerous tissue, which has been previously described for HCC (10.1021/acs.jproteome.8b00323).
For glycan 87, ions that locate fucose residue in the chitobiose core and in the outer arm of the structure have also been highlighted.
Reviewer 1. Finally, the comparison between serum vs tissue only based on glycome data does not provide any relevant information.
Our reply
Yes, that is one of the conclusions of the manuscript. The lack of correlation is a result. Observe that this contrasts with previous findings for hepatocellular carcinoma, in which authors state that “This work supports the hypothesis that the increased levels of fucosylated N-linked glycans in HCC serum are produced directly from the cancer tissue.” (10.1021/acs.jproteome.8b00323). We found no evidence, which is, therefore, a result.
Reviewer 1. The glycoproteome is complex and it is impossible to infer any potential cellular origin without the knowledge of the glycoprotein carrier.
Our reply
The authors agree with the above reviewer´s remark.

Round 3
Reviewer 1 Report
The authors have addressed the reviewer's questions.